# Disease-specific B cell clones are shared between patients with Crohn's disease

Prasanti Kotagiri [1,2,3,4] ✉, William M. Rae[1,2,5], Laura Bergamaschi[1,2], Diana Pombal[2], Ji-Yeun Lee[4], Nurulamin M. Noor [2], Raoul S. Sojwal[6], Samuel J. S. Rubin [6], Lukas W. Unger[1,2,7], Sofie H. Tolmeijer[2], Giulia Manferrari[2], Rachael J. M. Bashford-Rogers[2,8], David B. Bingham[4], Anton Stift[7], Stephan Rogalla[6], John Gubatan [6], James C. Lee[1,9], Kenneth G. C. Smith [1,2], Eoin F. McKinney [1,2], Scott D. Boyd [4] & Paul A. Lyons [1,2] ✉

B cells have important functions in gut homeostasis, and dysregulated B cell populations are frequently observed in patients with inflammatory bowel diseases, including both ulcerative colitis (UC) and Crohn's disease (CD). How these B cell perturbations contribute to disease remains largely unknown. Here, we perform deep sequencing of the B cell receptor (BCR) repertoire in four cohorts of patients with CD, together with healthy controls and patients with UC. We identify BCR clones that are shared between patients with CD but not found in healthy individuals nor in patients with UC, indicating CD-associated B cell immune responses. Shared clones are present in the inflamed gut mucosa, draining intestinal lymph nodes and blood, suggesting the presence of common CD-associated antigens that drive B cell responses in CD patients.

IBD is classically thought to be caused by aberrant T cell and innate immune responses, but mounting evidence additionally implicates antibody-mediated immunity in disease pathogenesis. However, the nature of this B cell involvement is unclear, especially whether it is causal or simply the result of a breach in the mucosal barrier[1–4].

Healthy gut mucosa is dominated by the IgA subclass of antibodies, which limits microbial infiltration[5]. In inflammation, however, other subclasses of antibodies become more abundant. In IBD, increased numbers of IgG+ B cells are detectable, as well as antibodies against gut commensals. Anti-commensal IgG may drive abnormal inflammatory responses with the ability to fix complement and activate other immune cells via fragment crystalisable (Fc)γ receptors. Indeed, the Fc receptor variant FCGR2A-R131, which has lower IgG binding affinity, is protective against UC, one of the two major types of

IBD[4,6]. In CD, the other major type of IBD, B cell clonal expansion and increased prevalence of the auto-reactive B cell receptor (BCR) VH4-34 are detectable in peripheral blood[7]. However, further work is needed to better understand the precise role in the disease of the clonal B cell antigen-specific responses observed in CD.

The BCR denotes the unique clonal identity of a B cell. Generation of the BCR occurs during B cell development in the bone marrow with rearrangement of the immunoglobulin receptor genes. This lineage then undergoes further diversification after antigenic stimulation via class-switching recombination (CSR) and somatic hypermutation (SHM) in secondary lymphoid organs[8]. The BCR repertoire refers to the range of individual BCRs that collectively provide the diversity of antigen receptors required to recognise any antigen. The hypervariable complementarity-determining region 3 (CDR3) of the BCR is

[1]Cambridge Institute of Therapeutic Immunology and Infectious Disease, Jeffrey Cheah Biomedical Centre, University of Cambridge, Cambridge, UK. [2]Department of Medicine, University of Cambridge School of Clinical Medicine, Cambridge, UK. [3]Department of Immunology and Pathology, Monash University, Melbourne, VIC, Australia. [4]Department of Pathology, Stanford University, Stanford, CA 94305, USA. [5]Discovery Sciences, AstraZeneca, Cambridge Biomedical Campus, Cambridge, UK. [6]Division of Gastroenterology and Hepatology, Stanford University, Stanford, CA 94305, USA. [7]Division of Visceral Surgery, Department of General Surgery, Medical University of Vienna, Vienna, Austria. [8]Present address: Department of Biochemistry, South Parks Road, University of Oxford, Oxford OX1 3QU, UK. [9]Present address: The Francis Crick Institute and UCL Institute of Liver and Digestive Health, Division of Medicine, Royal Free Campus, London, UK. ✉e-mail: pk488@cam.ac.uk; pal34@cam.ac.uk

formed by the combination of V, D and J genes, and its sequence (and consequently structure) is a key determinant of antigen binding. Identifying BCR sequences that are shared between different immune responses can, therefore, provide a powerful way of identifying B cell clones with shared antigen targets[9–11].

Here in this study, using analysis of the BCR repertoire from blood, gut mucosa, and lymph nodes (LNs), we develop a method to illustrate the presence of common antigens through the analysis of shared clones. Using this approach, we identify CD-associated clonal B cell populations within LNs near sites of inflammation that are shared between patients.

## Results

### Increased plasmablasts and altered gene expression in B cells of patients with CD

To determine whether peripheral blood immune phenotypes were associated with disease, we analysed B cell subsets in patients with active, untreated CD and HC using flow cytometry (Supplementary Fig. 1a, b and Supplementary Table 1- cohort 1). This revealed an increase in plasmablasts and a decrease in switched memory B cells in patients with CD (Fig. 1a). In keeping with increased plasmablast numbers, increases in total serum IgA, IgG and IgG1 titres were observed relative to HCs (Fig. 1b, Supplementary Fig. 1a, c and Supplementary Table 1- cohorts 1 and 2). To further assess the B cell differences, we performed bulk gene expression analysis of CD19⁺IgD⁺CD27⁻ and CD19⁺IgD⁻CD27⁺CD24⁺CD38⁻ B cells, purified from peripheral blood by fluorescence-activated cell sorting (FACS) (Supplementary Fig. 1a, d, Supplementary Table 1- cohort 1 and Data 1). Since CD19⁺IgD⁺CD27⁻ B cells are mostly naïve B cells with few antigen-experienced ones, we refer to them as naïve B cells. Similarly, since CD19⁺IgD⁻CD27⁺CD24⁺CD38⁻ cells are mainly memory B cells, we refer to them as memory B cells. Principal component and differential gene expression analyses of naive and memory B cells, comparing active CD with HC revealed clear differences in transcriptional states, suggesting that altered B cell responses may be involved in disease pathology (Fig. 1c–e). Volcano plots comparing gene expression profiles of CD and HC illustrated a total of 1168 upregulated and 1225 downregulated genes in the naive B cell transcriptome (Supplementary Data 2). Highly upregulated genes (log fold change >0.5) included *GBP1, 2, 4* and *5* and *ITPR1* representative of interferon stimulating pathways (MSigDB Hallmark, adjusted *P* value 0.028) and NOD-like receptors signalling pathways (KEGG 2021 Human, adjusted *P* value 0.022) whilst highly downregulated genes (log fold change <−0.5) included ribosomal proteins enriching for translation (KEGG 2021 Human, adjusted *P* value 5.71e-17). Type I IFNs suppress protein translation, in keeping with our findings[12]. A greater number of genes were dysregulated in the memory B cell transcriptome, with 1609 genes upregulated and 1243 genes downregulated (Supplementary Data 3). Highly upregulated genes included *TRAF1*, a key intracellular signalling molecule associated with auto-antibody formation, and downregulated genes included *NR4A1* and *EGR3*, both negative regulators of self-reactive B cells, the combination potentially contributing to autoimmunity[13–15].

To determine if the transcriptomic changes were unique to CD, we expanded our comparison to include patients with active, untreated UC. CD showed a greater number of differentially expressed genes than UC in both naive and memory B cells when compared to healthy donors (Fig. 1f, g). In total, 2393 and 650 genes were dysregulated in the naïve B cell transcriptomes of patients with CD and UC, respectively, while 2852 and 1374 genes were dysregulated in the memory B cell transcriptomes of patients with CD and UC, respectively. There was a large overlap of dysregulated genes shared between UC and CD, suggestive of a shared aberrant B cell transcriptional signature in IBD. In the naïve B cell transcriptome, 81% of the genes that were dysregulated in UC were also dysregulated in CD, while 80% of dysregulated genes in the memory B cell transcriptome were also shared between UC and CD (Fig. 1f, g). To further investigate this, we clustered IBD patients and HCs using differentially expressed genes from both the naive and memory B cell populations. Broadly, UC and CD patients clustered together. Naïve B cell transcriptomes of only a few IBD samples clustered with HC, and memory B cell transcriptomes were also distinct, with only a few HC samples clustered with IBD (Supplementary Fig 2a, b). Pathway analysis of statistically significant overlapping upregulated genes in the CD and UC naïve transcriptome when compared with health (*n* = 336 genes) revealed upregulation of the IFN alpha signalling pathway (BioCarta, adjusted *P* value 0.16), whilst downregulated genes (*n* = 193) represented ribosomal pathways (KEGG 2021 human adjusted *P* value 1.79e-6). Pathway analysis of statistically significant overlapping upregulated genes in the CD and UC memory transcriptome when compared with health (*n* = 609 genes) revealed upregulation of chromatin modifying enzymes (adjusted *P* value 6.0e-4), whilst downregulated genes (*n* = 497) were enriched for ribosomal pathways (KEGG 2021 human, adjusted *P* value 6.0e-79), similar to the findings in naïve B cells.

### Evidence for common antigens in Crohn's disease through analysis of B cell clonal sharing

To address whether there might be common antigens involved in driving CD, we investigated whether shared clones were present within the BCR repertoire (Supplementary Fig. 1a and Supplementary Data 4, 5). Conventionally, clones −as defined by their CDR3 nucleotide sequence−are considered shared if they contain matching V and J genes with an identical CDR-H3 length and at least 85% amino acid CDR-H3 sequence homology (Fig. 2a). We calculated the number of shared B cell clones pairwise within each disease/tissue group. A high degree of clonal overlap suggests responses to shared epitopes between patients. In keeping with the expected high diversity of naive B cells, the predominant B cell subset in peripheral blood, there was minimal clonal sharing (Supplementary Table 1- cohort 2). However, considerable clonal sharing was observed among repertoires within LNs (Supplementary Table 1- cohort 3) close to areas of inflammation, which had been resected intra-operatively in patients with severe CD (Fig. 2b). Sharing was greatest in antigen-experienced clones (IgA+ and IgG+) (Fig. 2c), suggesting that B cell responses to shared antigens are common amongst CD patients. To further assess the nature of these clones, we selected those that were both shared and disease-associated (absent in control post-mortem mesenteric LN, MSLN)[16] (Fig. 2a). Using this method, we identified 12,108 unique clones that were present in at least two individuals with CD and that were absent in the post-mortem MSLN samples. These CD-associated clones were enriched in all immunoglobulin isotypes, but with a greater proportion in class-switched isotypes (e.g., IgG+, IgA+), consistent with a T cell-dependent B cell response (Fig. 2d). The top 20 most frequently identified convergent clonotypes were found in 20 or more patients (Fig. 2e) and had preferential use of VH4. V gene use in the 12,108 clones revealed that IGHV4-39, 4-4, 4-59, 3-23, and 3-7 were the five most expanded V genes, representing 51% of shared clones (Supplementary Fig. 3a, b). This oligoclonal expansion of CD-associated B cell clones provides further support for potential common antigens associated with this disease.

Antibody-secreting cells generated in Peyer's patches and MSLN do not immediately localise to gut tissue but recirculate systemically before returning to the gut via expression of gut-specific homing receptors, including MAdCAM, CCR9 and the α4β7 integrin[17]. To test the reproducibility of the CD-associated clonotype findings in our LN analysis, we investigated the BCR repertoire derived from PBMCs of patients with CD (Supplementary Table 1- cohort 2) and found the same CD-associated clones observed in lymph nodes (Supplementary Table 1- cohort 3) enriched significantly in CD compared to HCs (Fig. 2f)[7]. This cohort of patients with active CD were not on steroids or

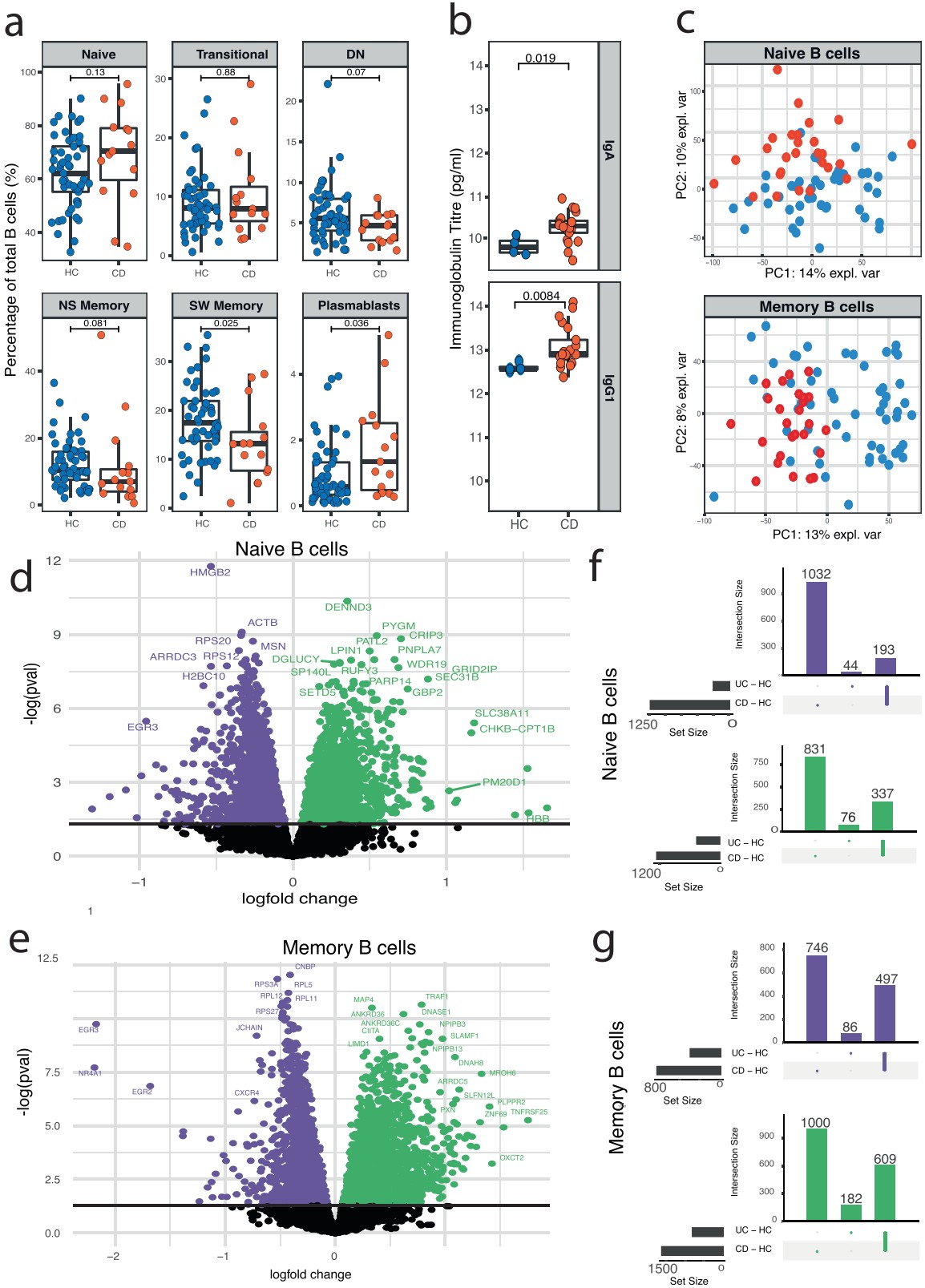

biologics (tumour necrosis factor antagonist therapies, interleukin-12/23 inhibitors, or integrin inhibitors) at the time of sampling. We also observed significantly more convergence of CD-associated clones in CD when analyzed by isotype, particularly with higher frequencies in the IGHM, IGHA1/A2, and IGHG1/2 isotypes (Supplementary Fig. 3c).

Similarly, shared clones derived from the CD PBMC repertoire (present in 2 or more individuals with CD and absent from health,

$n = 765$ clones) had greater enrichment in CD LN compared with MSLN (Supplementary Fig. 4a, b). A comparison of sequence lengths revealed that shared clones originating from LN were significantly longer with a mean length of 16.5 amino acids, compared with those derived from PBMCs with a mean length of 15.5 amino acids (Supplementary Fig. 4c). A comparison of J gene frequency in shared clones derived from LN versus PBMCs yielded a Pearson's chi-squared test $P = 0.0004998$,

**Fig. 1 | Increased plasmablasts and altered gene expression in B cells of patients with Crohn's disease. a** B cell immunophenotyping. Boxplots showing proportions of B cell populations, by disease group. Unpaired two-sided Wilcoxon's signed-rank test. Naïve (CD19+IgD+CD27−), double-negative (DN) B cells (CD19+IgD−CD27−), non-switched (NS) memory (CD19+CD27+IgD+), switched (SW) memory (CD19+IgD−CD27+) and plasmablasts (CD19−/lowCD20−CD27+CD24−CD38+) and transitional B cells (CD19+IgD+CD27− CD24+ CD38+). Healthy individuals (*n* = 53) and CD (*n* = 15). HC coloured in blue and CD coloured in red. **b** Immunoglobulin titres. Boxplot of immunoglobulin titres split according to isotype and disease. Immunoglobulin titres in healthy individuals (*n* = 4), patients with CD (*n* = 20). Unpaired two-sided *t*-test. HC coloured in blue and CD coloured in red. **c** Assessment of transcriptional differences in B cell**s**. PCA plot of naïve and memory B cells comparing CD with HC. HC coloured in blue and CD coloured in red. **d** Differential gene expression in naïve B cells. Volcano plot representing differentially expressed genes between HC and CD (Benjamini–Hochberg correction, FDR <0.05). Upregulated genes in green and downregulated genes in purple. *p* values and FDRs are determined by moderated *t*-tests and multiple test corrections using limma.
**e** Differential gene expression in memory B cells. Volcano plot representing differentially expressed genes between HC and CD (Benjamini–Hochberg correction, FDR <0.05). Upregulated genes in green and downregulated genes in purple.

*p* values and FDRs are determined by moderated *t*-tests and multiple test corrections using limma. **f** Differential gene expression in naive B cells in IBD. Interaction plot illustrating differentially expressed genes in CD and UC compared with health. Upregulated genes in green and downregulated genes in purple. The gene distribution is indicated by the lines and dots with the number of genes sharing that disease group distribution indicated by the vertical histogram bars. The total number of upregulated/downregulated genes identified in each disease group when compared with health is indicated in the histogram to the left of the plot. **g** Differential gene expression in memory B cells in IBD. Interaction plot illustrating differentially expressed genes in CD and UC compared with health. Upregulated genes in green and downregulated genes in purple. The gene distribution is indicated by the lines and dots with the number of genes sharing that disease group distribution indicated by the vertical histogram bars. The total number of upregulated/downregulated genes identified in each disease group when compared with health is indicated in the histogram to the left of the plot. For **c**–**g**, naïve B cells (HC = 49, CD = 24 and UC = 32) and memory B cells (HC = 54, CD = 26 and UC = 26). Boxplots show the median (centre line) and interquartile range (25th–75th percentile of the data) and whiskers indicate range of data 1.5 times the interquartile below and above the 25th and 75th percentile, respectively.

---

indicative of differences in J gene usage across the two sites. A post-hoc Bonferroni-corrected pairwise analysis revealed significant proportional differences between IGHJ4 (PBMCs 84%, LN 77%) versus IGHJ5 (PBMCs 16%, LN 23%) with a corrected *P* value = 0.004, and IGHJ5 (PBMCs 41%, LN 54%) versus IGHJ6 (PBMCs 59%, LN 46%) with a corrected *P* value = 0.006 (Supplementary Fig. 4d). The enrichment of clones underscored the reproducibility of these convergent B cell responses in CD disease mechanisms.

We further refined the number of CD-associated clones derived from LNs by focusing only on the top 3000 expanded clones per person, hypothesising that expanded clones in the setting of active disease are more likely to be relevant to pathogenesis. Selecting those that were present in two or more individuals and absent in the postmortem samples reduced the number of CD-associated LN clones to 4698 from 12,108 clones. With this refined clone list, we similarly found greater enrichment of CD-associated LN clones in PBMCs in both the non-switched and switched repertoire (Supplementary Table 1- cohort 2) (Supplementary Fig. 4e).

**Antigen-driven clonal expansion of plasmablasts in active CD**
To confirm that the increase in plasmablasts in peripheral blood (Fig. 1a) was driven at least in part by a response to shared antigens, we generated BCR repertoires from plasmablasts isolated during active disease (Supplementary Fig. 1c and Supplementary Table 1- cohort 1). Plasmablasts are short-lived and, therefore, are expected to be enriched for clones/sequences that are relevant to the current inflammatory response. Similar to the previous validation cohort (Supplementary Table 1- cohort 2), this group of patients had active disease and were not on concurrent immunosuppression. We found that the CD-associated clones were also present in CD plasmablasts and were more prevalent there compared to CD PBMC. These clones were not as prevalent in the plasmablasts of healthy individuals or patients with UC, confirming their association with CD. This was especially true when testing for enrichment of the refined clonal list (4698 clones) where this finding held in both the non-switched and class-switched repertoire (Fig. 2g and Supplementary Fig. 5a). While no single clone was shared between all CD patients which may be reflective of disease subtypes by antigenic target or a polyclonal response to the same target, CD-associated clones were shared between multiple cases with convergence in IGHM, IGHA1 and IGHG2 isotypes (Supplementary Fig. 5b).

Using the same clustering technique that was applied to identify CD-associated clones, we identified shared clones in the UC plasmablast repertoire in keeping with recent findings[9]. We selected clones

that were shared in UC patients and absent in healthy controls. To test if these clones were UC-associated, we assessed the level of enrichment in CD. These UC-associated clones were not enriched in patients with CD any more than in idiopathic pulmonary fibrosis, a disease control not associated with IBD, further highlighting the difference in B cell antigenic responses in CD and UC (Fig. 2h).

**Increased affinity maturation in CD-associated clones indicates antigen-targeted immunity**
In IBD, we observed, a global reduction in SHM in circulating plasmablasts across all class-switched isotypes when compared to healthy controls (Fig. 3a), as has been reported following infection[10,11]. There was also a significantly increased fraction of unmutated/lowly mutated clones (<1 % SHM in the IGHV gene) (Fig. 3a). This decrease in SHM may be attributable to an extrafollicular response or recently recruited clones from the naïve repertoire expanding in the germinal centre. The presence of well-formed germinal centres in CD LN is more supportive of the latter but does not exclude the possibility of a concurrent extrafollicular response or plasmablast generation via tertiary lymphoid structures (Fig. 3b).

We then compared the SHM of CD-associated and non-CD-associated clones within the plasmablast repertoire of patients. This revealed that CD-associated clones had increased SHM across all isotypes (Fig. 3c), indicative of affinity maturation and providing further evidence of antigen-targeted immunity.

**Serum antibodies against *Klebsiella* species distinguish CD from UC and HC**
We profiled antibodies present in the serum generated by plasmablasts in circulation and long-lived plasma cells in the bone-marrow. We assessed the differential abundance of gut associated pathogen-binding antibodies in patients with CD compared to HC and those with UC combined, to obtain a disease specific response (Supplementary Table 1 - cohorts 1 and 2). Upregulation of serum antibodies targeting *Klebsiella* species was a dominant feature of patients with CD (Fig. 4a and Supplementary Data 6).

We used a sparse partial least-squares discriminant analysis (sPLS-DA) to determine which pathogen-associated antibodies were most informative for disease prediction. CD could be discriminated from UC and HC combined based on 20 key antibody populations selected by the model (Fig. 4b). The area under the receiver operator characteristic (AUROC) curve for patient cluster classification based on these 20 antibodies was 0.97 (97% chance of accurate cluster prediction). This suggests that antibodies play a role in disease (Fig. 4c).

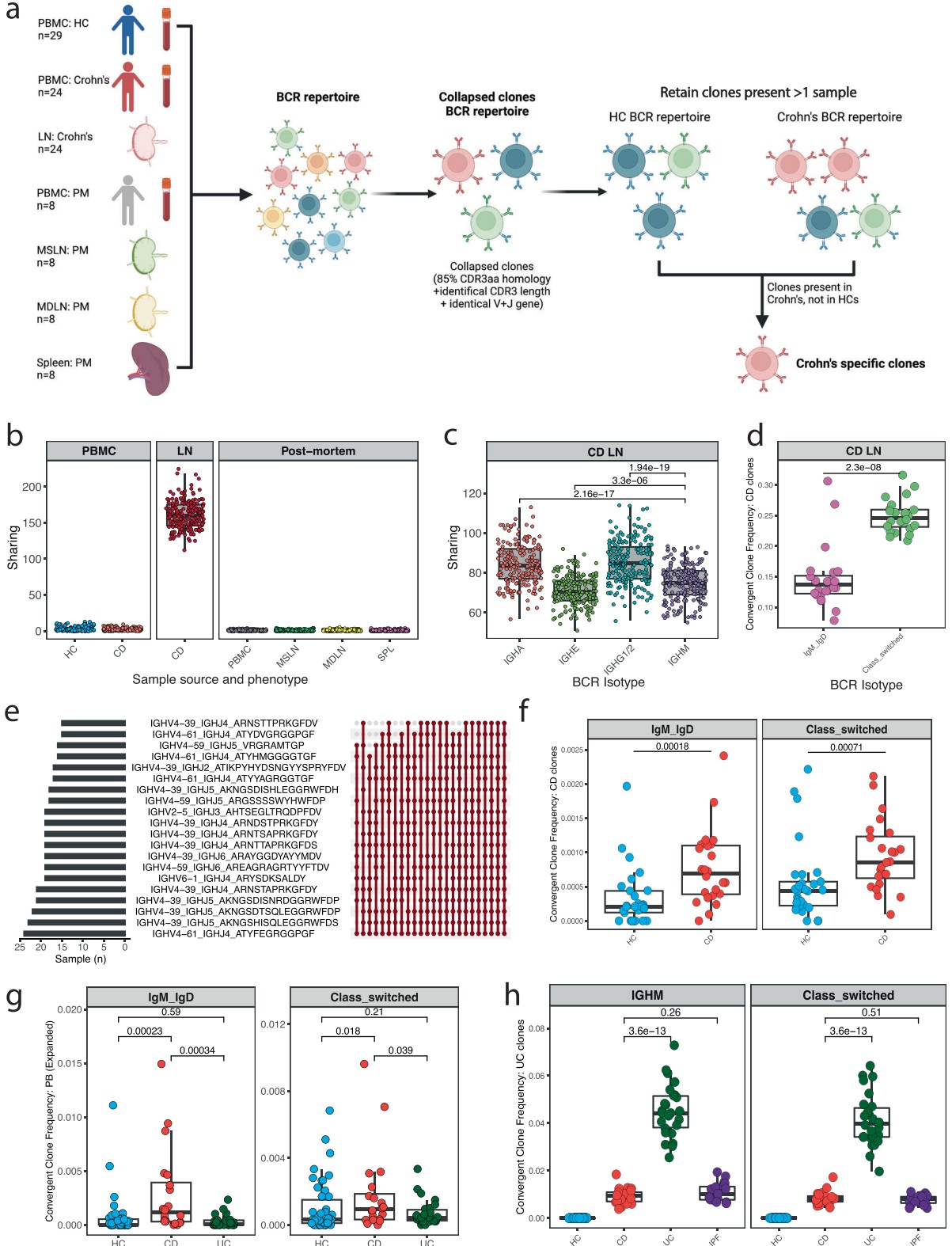

## Intestinal BCR repertoire analysis reveals enrichment of CD-associated clones

To gain a deeper understanding of the intestinal-systemic interface, we investigated the bulk BCR repertoire in intestinal mucosa during active disease (Supplementary Table 1- cohort 4). We compared the repertoires obtained from areas of active inflammation (CD $n = 27$, UC $n = 19$) and matched less inflamed samples from the same subjects (CD

$n = 28$, UC $n = 19$) with non-IBD uninflamed controls ($n = 12$) (Fig. 5a and Supplementary Data 5, 7).

In light of elevated serum IgA and G observed in IBD, we wanted to assess if there were differences in sub-isotype expression. We assessed the proportion of unique B cell clones within IgA and G subclasses, counting each unique VDJ region only once to ensure results were not skewed by the differential

**Fig. 2 | Evidence for common antigens in Crohn's disease through analysis of B cell clonal sharing. a** Schematic of shared clones. Peripheral blood mononuclear cells (PBMC), lymph nodes (LN), healthy controls (HC), mediastinal lymph nodes (MDLN), mesenteric lymph nodes (MSLN), post-mortem (PM). Created in BioRender. Kotagiri, P. (2024) BioRender.com/c85s966. **b** Assessment of clonal sharing. Randomly selected 8 patients per group and 1500 unique clones per patient (200 iterations). Boxplots of the number of clones shared in at least two patients split according to group. Each dot represents an iteration. **c** Isotype-specific clonal sharing in CD LN. Randomly selected 16 patients per isotype and 500 unique clones per patient (200 iterations). Boxplots of the number of clones shared in at least two patients split according to isotype. Each dot represents an iteration. Unpaired two-sided Wilcoxon's signed-rank test. **d** Identified potentially CD-associated LN clones. Clones were identified based on the presence of two or more CD-inflamed LN samples and the absence of MSLN from the post-mortem cohort. Boxplot of convergence in IGHM-IGHD and class-switched clones in CD LN BCR repertoire. Each dot represents a sample. Unpaired two-sided Wilcoxon's signed-rank test. $n = 24$. **e** Dominant clones: 20 most frequent CD clones identified in LN. Horizontal bars present the number of LN samples a given clone is present in. Lines connecting dots on the right illustrate the pattern of co-existence of multiple of clones in a given

sample. The dot symbolises the presence of the clone in the combination. **f** Validation of CD-associated LN clones in PBMCs. Boxplots of enrichment. Each dot represents a sample. Unpaired one-sided Wilcoxon's signed-rank test. Healthy individuals ($n = 29$) and CD ($n = 24$). **g** Validation of CD-associated LN clones in plasmablasts. Assessed clonal convergence of the list of refined CD-associated clones derived from LN in plasmablasts according to unique UMI and not unique clone, thus taking clonal expansion into consideration. Boxplot representing convergence split according to isotype and disease. Each dot represents a sample. Unpaired one-sided Wilcoxon's signed-rank test. **h** Shared clones in UC. UC-associated clones were identified based on the presence in two or more UC plasmablast samples and absent in health. Boxplot of convergence in IGHM and class-switched clones in HC, UC, CD and Idiopathic pulmonary fibrosis (IPF) BCR repertoire. Each dot represents a sample. Unpaired two-sided Wilcoxon's signed-rank test. For **g**, **h** $n = 46$ for healthy individuals and $n = 20$, $n = 26$ and $n = 19$, for patients with CD, UC and IPF, respectively. Boxplots show the median (centre line) and interquartile range (25th–75th percentile of the data) and whiskers indicate range of data 1.5 times the interquartile below and above the 25th and 75th percentile, respectively.

mRNA content of B cell subsets. This revealed no significant difference in sub-isotype use between disease groups (Supplementary Fig. 6a, b).

SHM changes were consistent with findings in circulating plasmablasts with a global reduction in the class-switched isotypes including IGHA1 and 2, and IGHG1, 2 and 3 (Fig. 5b). This finding was most pronounced in UC as observed in the circulating plasmablasts (Fig. 3a). There was no significant difference when comparing paired inflamed and uninflamed regions, indicating a global aberrancy in B cells in the mucosa of patients with IBD independent of inflammatory state (Fig. 5b). We next examined the contribution of various VH genes to the repertoire. Compared with non-IBD controls, both UC and CD showed decreased IGHV4-61 use and increased IGHV1-8 use across non-switched and class-switched isotypes. Inflamed and uninflamed regions showed similar V gene usage profiles (Fig. 5c, Supplementary Fig. 6c and Supplementary Data 8).

Assessing for CD-associated clones in these samples revealed greater enrichment of CD-associated clones in the repertoire of patients with CD compared to HC or patients with UC, using both the expansive and refined list of clones with significant enrichment in the IGHM and IGHA2, as well as a trend towards enrichment in IGHA1 and the IGHG classes (Fig. 5d and Supplementary Fig. 6d, e). There was no greater increase in clonal convergence when comparing the inflamed and uninflamed regions in patients with CD or when comparing different regions of inflamed bowel (Supplementary Fig. 6f-g). In keeping with the findings in circulating plasmablasts, CD-associated clones detected in CD gut mucosa had increased rates of SHM compared to clones that were deemed non-CD-associated, which suggests affinity maturation (Fig. 5e).

## Discussion

This study of the BCR repertoire in intestinal lymph nodes (LNs), mucosa, and blood demonstrates the presence of CD-associated clones shared among multiple patients. These clones are found in LNs, intestinal mucosa and PBMCs during active disease, but are not enriched to the same extent in healthy controls or patients with active UC. These shared clones are more commonly found in the IGHM, IGHA and IGHG repertoire of both blood and intestinal mucosa.

While neither the antigenic specificity nor the neutralising capacity of antibodies encoded by identified BCR sequences has been determined experimentally, their convergence between multiple patients suggests the presence of common, disease-associated antigens rather than a non-specific polyclonal activation of B cells. The global reduction in SHM we observed in the plasmablasts and intestinal mucosa of patients with CD suggests polyclonal

stimulation of B cells bearing more germline repertoires, whilst the increased SHM of CD-associated clones is likely representative of antigen selection occurring. These disease-associated clones are unlikely to be driven by treatment as we can validate their presence in patients with active CD, not on immunosuppression. However, further validation is needed in a cohort of active untreated CD patients with characterisation of LN-derived BCR repertoires is performed.

CD-associated clones are more commonly found in the IgA1, IgA2 and IgG2 CD BCR repertoire despite IgA and IgG1 serum antibody levels being elevated. The lack of correlation between the BCR repertoire and serum IgG1 titres may be due to differences in both cellular and immunoglobulin kinetics. "Steady state" serum IgG immunoglobulin is made predominantly by long-lived plasma cells in the bone marrow and thus may not correlate with the BCR repertoire of short-lived plasmablasts. An additional explanation could be found in differences in isotype BCR sequencing depth, that could impact the identification of shared clones.

In keeping with the presence of CD-associated clones, we uncovered a CD-specific antibody profile. We identified a robust antibody response against *Klebsiella Pneumonia, Bacillus circulans, Lactobacillus species*, and *Enterococcus faecalis* in patients with CD. *Bacillus circulans* and *Lactobacillus* species are likely targeted due to their flagellar components. Anti-flagellin antibodies have previously been associated with ileal and fibrostenotic CD[18]. In addition, gut-derived *Enterococcus faecium* from IBD patients has been shown to induce colitis in genetically susceptible mice[19], whilst *Klebsiella* has been associated with exacerbations and increased severity of IBD, in addition to causing intestinal inflammation in colitis-prone mice[20].

Antibodies in patients with IBD have been previously shown to target both self-antigens and microbial components. UC commonly exhibits a high expression of the self-reactive antibody pANCA[21], which targets nuclear histone 1 of polymorphonuclear leucocytes. In contrast, CD is characterised by elevated levels of anti-microbial antibodies, including ASCA (*anti-Saccharomyces cerevisiae* antibody)[22] targeting mannan in the cell wall of baker's yeast, anti-OmpC targeting the outer-membrane protein OmpC of *Escherichia coli*[21], anti-flagellin targeting bacterial flagellins[23,24], and anti-I2 targeting a component of *Pseudomonas fluorescens*[25,26]. Our work further expands the microbial targets of CD. However, it remains unclear whether all of these detected antibodies play a direct role in the pathogenesis of IBD or if they merely serve as markers of an aberrant immune response. The increased frequency of antibodies against pathobionts is suggestive of a pathogenic role. Going forward, the

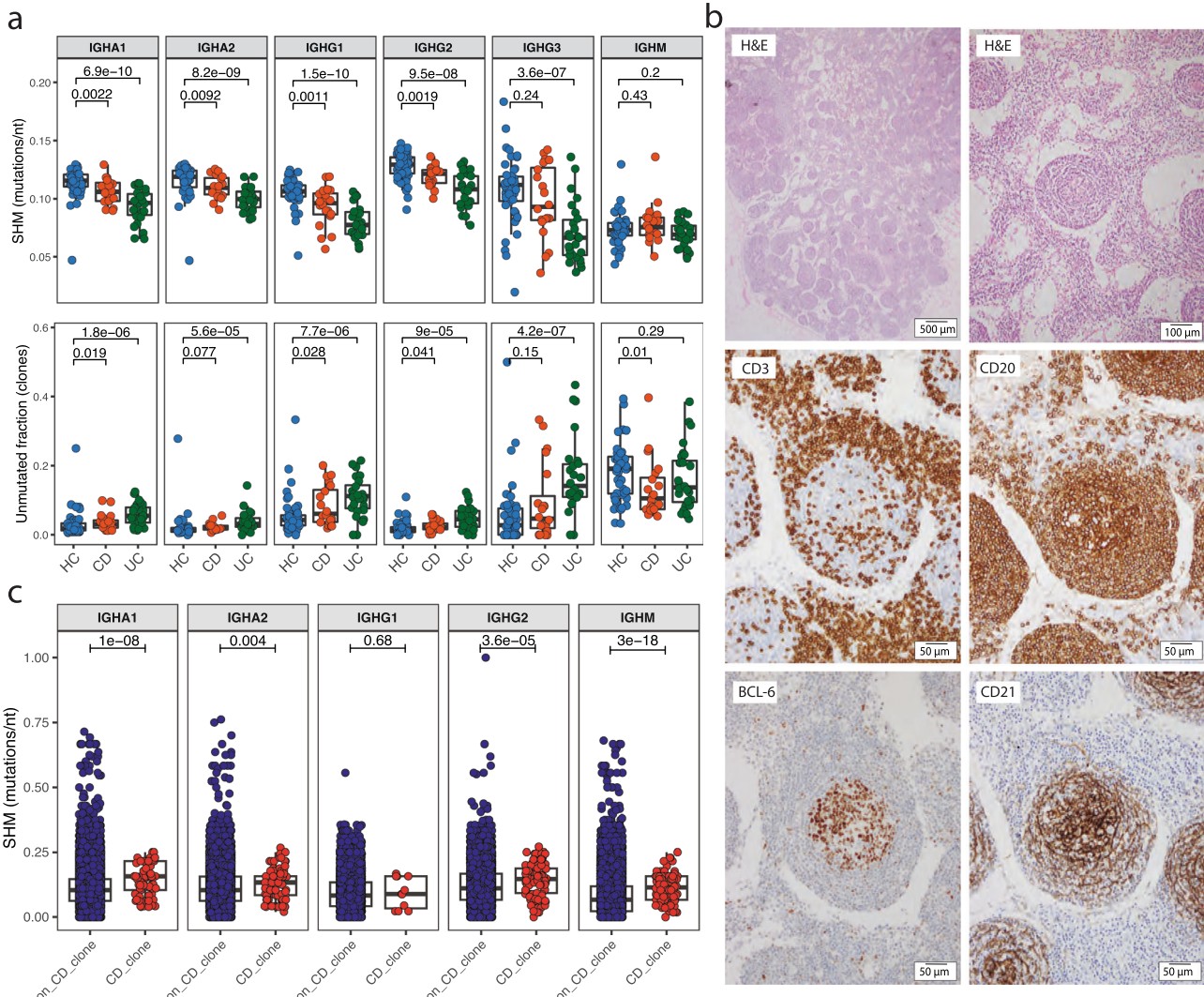

**Fig. 3 | Antigen-driven clonal expansion of plasmablasts in active Crohn's disease. a** SHM. BCR repertoire in plasmablasts (CD19⁺IgD⁻CD27⁺CD24⁻CD38⁺). The top row of boxplots depicts SHM split according to isotype and disease, averaged per sample and isotype. Each dot represents a sample. Unpaired two-sided Wilcoxon's signed-rank test. The bottom row of boxplots assesses the proportion of lowly mutated clones, defined as SHM <1% across V region averaged per sample and isotype. Boxplot split according to isotype and disease. Each dot represents a sample. Unpaired two-sided Wilcoxon's signed-rank test. **b** Germinal centre histology. Representative immunohistochemistry of germinal centres within an MSLN draining an inflamed region of the bowel during an acute flare of CD requiring surgical resection using CD3 (T cells), CD20 (B cells), BCL6 (GC B cells (or T follicular helper cells) and CD21 (follicular dendritic cells). **c** SHM of shared clones. Boxplot of SHM comparing non-CD and CD-associated clones in CD patients split according to isotype. Each dot represents a unique clone per patient. Unpaired two-sided Wilcoxon's signed-rank test. For **a**, **c**, n = 46 for healthy individuals and n = 20 and n = 26 for patients with CD and UC, respectively. Boxplots show the median (centre line) and interquartile range (25th–75th percentile of the data) and whiskers indicate range of data 1.5 times the interquartile below and above the 25th and 75th percentile, respectively.

generation of monoclonal antibodies from convergent IGH sequences would allow their further functional characterisation of the identified CD-associated clones.

Taken together, our data further supports the role of pathogenic B cells in CD and also offers an opportunity, provided epitope-specific antibodies can be fully elucidated, to identify additional antigenic targets. This identification may prove valuable in aiding diagnosis, discriminating between disease endotypes, and defining novel treatment targets.

## Methods
### External datasets
Public datasets used in the study include Crohn's disease and control samples from ref. 7 generated in our laboratory and post-mortem samples from ref. 16

### Ethics
Ethical approval was obtained from the Medical University of Vienna's Institutional Review Board (EK number: 1480/2016), from the Cambridgeshire Regional Ethics Committee (REC08/H0306/21, REC08/H0308/176) and from Stanford University (Stanford Inflammatory Bowel Disease Immune Repertoire IRB 64710). All patients provided written informed consent.

### Peripheral blood mononuclear cell preparation and immuno-phenotyping by flow cytometry
Each participant provided 100 mL of peripheral venous blood collected into a 9 mL sodium citrate tube. Peripheral blood mononuclear cells (PBMCs) were isolated using Leucosep tubes (Greiner Bio-One) with Histopaque 1077 (Sigma) by centrifugation at 800×g for 15 min at room temperature. PBMCs at the interface were collected, rinsed twice

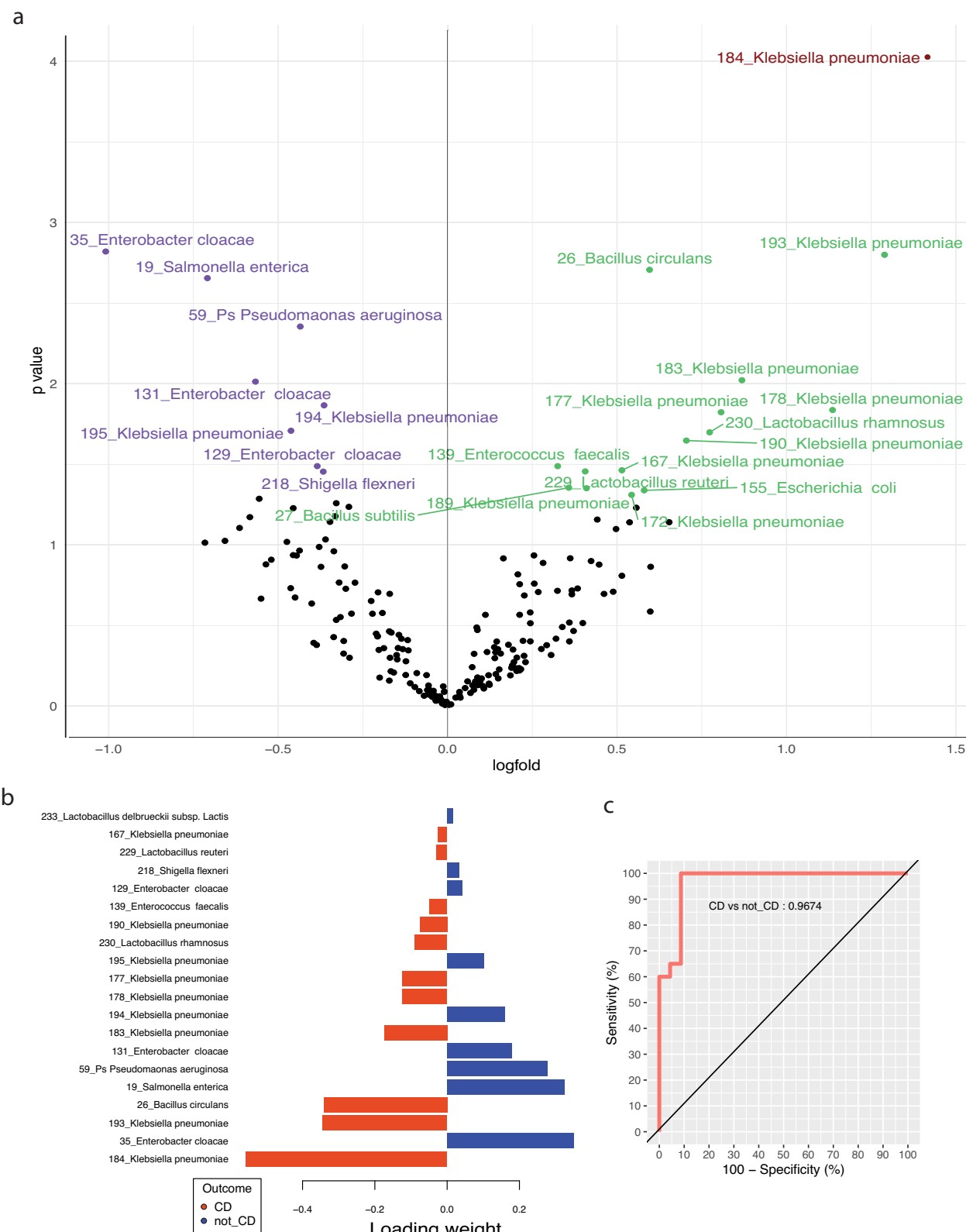

**Fig. 4 | Serum antibodies against *Klebsiella* species distinguish Crohn's disease from ulcerative colitis and healthy controls. a** Assessment of pathogen binding differences in serum of patients with IBD and in HC. Volcano plot representing differential pathogen-specific antibody abundance between CD and HC and UC combined. Upregulated antibodies in green and downregulated antibodies in purple. Significantly differentially expressed antibodies in burgundy. *P* values and FDRs are determined by moderated *t*-tests and Benjamini–Hochberg multiple test correction using limma. **b** Model selection. Twenty pathogens selected by sPLS-DA as most informative in predictive models discriminating CD from HC and UC combined. Bars indicate the loading coefficient weights of selected features (ranked from most to least informative in cluster prediction, from bottom to top). **c** Group prediction. AUROC curve showing sensitivity and specificity of group prediction, based on the 20 pathogens selected. For **a**–**c**, HC = 11, CD = 22 and UC = 9.

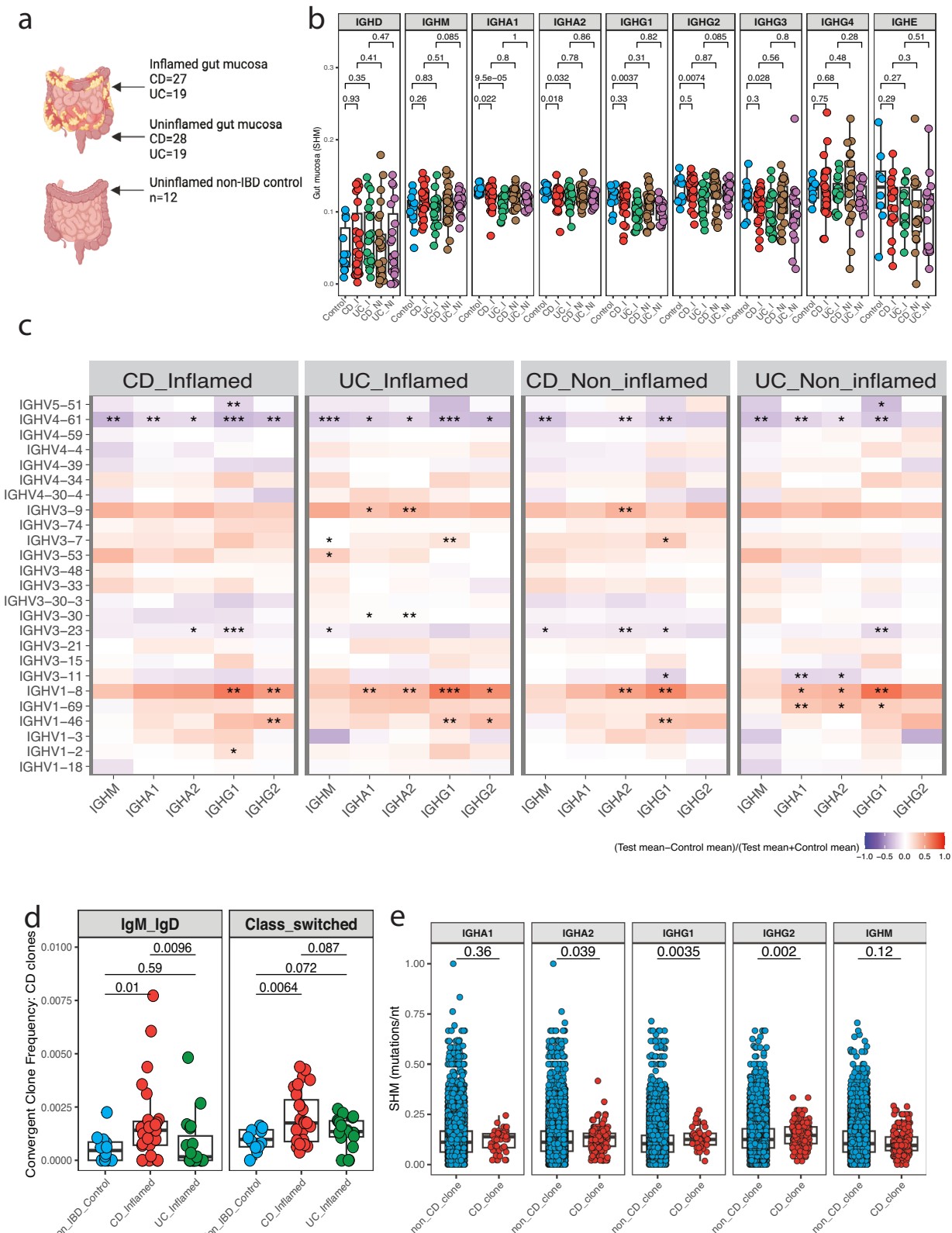

with autoMACS running buffer (Miltenyi Biotech) and cryopreserved in FBS with 10% DMSO. All samples were processed within 4 h of collection. Samples were stored at 4 °C and acquired within 4 h using a 5-laser BD Symphony X-50 flow cytometer. Single colour compensation tubes (BD CompBeads) or cells were prepared for each of the fluorophores used and acquired at the start of each flow cytometer run (Supplementary Table 2). For direct enumeration of B cells,

an aliquot of whole blood (50 μl) was added to BD TruCount tubes with 20 μl- BD Multitest 6-colour TBNK reagent (BD Biosciences) and processed as per the manufacturer's instructions. Samples were gated in FlowJo v10.2 using the following B cells markers Naive (CD19$^+$IgD$^+$CD27$^-$), double-negative B cells (DN) (CD19$^+$IgD$^-$CD27$^-$), non-switched (NS) memory (CD19$^+$CD27$^+$CD24$^+$CD38$^+$), memory (switched) (CD19$^+$IgD$^-$CD27$^+$CD24$^+$CD38$^+$) and plasmablasts

**Fig. 5 | Intestinal BCR repertoire analysis reveals enrichment of Crohn's disease-associated clones. a** Schematic of study participants and sample numbers. BCR repertoires were obtained from areas of active inflammation (CD $n = 27$, UC $n = 19$) and matched less inflamed samples from the same subjects and non-IBD uninflamed controls ($n = 12$). Created in BioRender. Kotagiri, P. (2024) BioRender.com/z51l727. **b** SHM. BCR repertoire in the gut mucosa. Boxplots depict SHM split according to isotype and inflammation status averaged per sample and isotype. Each dot represents a sample. Unpaired two-sided Wilcoxon's signed-rank test. Non-IBD control (Control), CD-inflamed (CD-I), UC-inflamed (UC-I), CD-uninflamed (CD-NI), UC-uninflamed (UC-NI). **c** V gene usage. Heatmap showing the difference between V gene proportion between CD and UC and controls. Difference calculated using the following method: (mean V gene proportion of disease − mean V gene proportion of HC)/(mean V gene proportion of disease + mean V gene proportion of controls). Unpaired two-sided Wilcoxon's signed-rank test FDR adjusted $p$ value (Benjamini–Hochberg): *$p < 0.1$, **$p < 0.05$, ***$p < 0.005$. **d** Validation of CD-associated LN clones in gut mucosa. Boxplots of enrichment. Each dot represents a sample. Unpaired one-sided Wilcoxon's signed-rank test. **e** SHM of shared clones. Boxplot of SHM comparing non-CD and CD-associated clones in inflamed mucosa of CD patients, split according to isotype. Each dot represents a unique clone per patient. Unpaired two-sided Wilcoxon's signed-rank test. For **b**–**e**, Non-IBD control = 12, CD-inflamed = 27, UC-inflamed = 19, CD-uninflamed = 28, UC-uninflamed = 19. Boxplots show the median (centre line) and interquartile range (25th–75th percentile of the data) and whiskers indicate range of data 1.5 times the interquartile below and above the 25th and 75th percentile, respectively.

($CD19^-CD20^-CD27^+CD24^+CD27^+CD38^+$). The number of cells falling within each gate was recorded. Plasmablasts were sorted ($CD19^+CD38^+IgD^-CD27^+CD24^-$) fresh into lysis buffer, and then the lysed cells were frozen.

### Serology
IgA and IgG1 levels in patient serum were measured using a ProcartaPlex immunoassay kit (Thermo Fisher) using 25ul of serum from each individual and run on a Luminex xMAP analyser. Raw data (MFI) were normalised to a concurrently measured seven point standard curve according to the manufacturer's instructions to return an absolute quantification (pg/ml). All measured values were encompassed by the standard distribution.

### Pathogen array
To detect IgG antibodies against specific bacterial and fungal strains, a custom pathogen microarray developed by the Austrian Institute of Technology (AIT) was used. This microarray includes lysates from 254 bacterial and six fungal strains sourced from sepsis patients, point-of-care systems detection projects and type strains from strain banks.

To prepare the lysates for the pathogen array, bacteria and fungi were cultivated in suitable culture media under specific conditions. The cultures were then pelleted through centrifugation, and the resulting pellets were washed and resuspended in an elution buffer. This eluate was processed with bead beating and filtered to obtain cell lysates.

For antibody profiling on the pathogen array, IgG was isolated from serum samples and standardised to 0.2 mg/100 μL. The microarrays were blocked, washed, and incubated with the sample mix. Detection was carried out using Alexa Fluor® 647 Goat Anti-Human IgG. Image acquisition and fluorescence intensity extraction were performed using the Tecan PowerScanner™ and GenePix Pro7 software. Median fluorescence intensities were used for bioinformatic analysis.

### Bulk RNA sequencing and analysis
RNA-Sequencing libraries were generated using the SMARTer® Stranded Total RNA-Seq v2 - Pico Input Mammalian kit (Takara) using 10 ng RNA as input following the manufacturer's protocol. Libraries were pooled together ($n = 96$) and sequenced using 75 bp paired-end chemistry across four lanes of a Hiseq4000 instrument (Illumina) to achieve 10 million reads per sample. Read quality was assessed using FastQC v.0.11.8 (Babraham Bioinformatics, UK), and SMARTer adaptors trimmed and residual rRNA reads depleted in silico using Trim_galore v.0.6.4 (Babraham Bioinformatics, UK) and BBSplit (BBMap v.38.67(BBMap - Bushnell B. - https://sourceforge.net/projects/bbmap/)), respectively. Alignment was performed using HISAT2 v.2.1.0[27] against the GRCh38 genome, achieving a greater than 95% alignment rate. Count matrices were generated using featureCounts (Rsubreads package[28] - and stored as a DGEList object (EdgeR package[29]) for further analysis. All downstream data handling was performed in R (R Core Team, 2015). Counts were filtered using filterByExpr (EdgeR) with a gene count threshold of 10 CPM and the minimum number of samples set at the size of the smallest disease group. Library counts were normalised using calcNormFactors (EdgeR) using the method 'weighted trimmed mean of M-values'. The function 'voom'[30] was applied to the data to estimate the mean-variance relationship, allowing adjustment for heteroscedasticity.

### BCR sequencing: lymph node and peripheral blood
LN tissue post-sampling was stored in liquid nitrogen. Plasmablasts were flow sorted using the following markers: $CD19^+CD38^+IgD^-CD27^+CD24^-$ and stored in RLT. RNA was extracted using Qiagen RNeasy Kits.

B cell receptor repertoire libraries were generated using the protocol described by ref. 7. Briefly 200 ng of total RNA extracted from tissue or 1.5 ng of RNA extracted from plasmablasts (14ul volume) was combined with 1uL 10 mM dNTP and 10uM reverse primer mix (2uL) and incubated for 5 min at 70 °C. The reverse primer mix used in the LN BCR repertoire generation were from ref. 7 whilst for the plasmablasts from ref. 31. The mixture was immediately placed on ice for 1 minute and then subsequently combined with 1 uL DTT (0.1 M), 1 uL SuperScriptIV (Thermo Fisher Scientific), 4 ul SSIV Buffer (Thermo Fisher Scientific) and 1 uL RNAse inhibitor. The solution was incubated at 50 °C for 60 min followed by 15 min inactivation at 70 °C. cDNA was cleaned with AMPure XP beads and PCR-amplified with a 5′ V-gene multiplex primer mix and 3′ universal reverse primer using the KAPA protocol and the following thermal cycling conditions: 1cycle (95 °C, 5 min); 5cycles (98 °C, 20 s; 72 °C, 30 s); 5cycles (98 °C, 15 s; 65 °C, 30 s; 72 °C, 30 s); 19cycles (98 °C, 15 s; 60 °C, 30 s; 72 °C, 30 s); 1 step (72 °C, 5 min). Sequencing libraries were prepared using Illumina protocols and sequenced using 250-bp paired-end sequencing on a MiSeq system (Illumina).

### BCR sequencing: gut mucosa
Colonic mucosal biopsies were obtained with forceps as part of the standard of care restaging colonoscopy for patients with inflammatory bowel disease or colon cancer screening for non-IBD healthy controls. Histologic inflammation was graded by a pathologist. RNA from mucosal biopsies was isolated Using Ambion's ToTALLY RNA™ RNA Isolation Kit. Complementary DNA (cDNA) was generated from total RNA using SuperScript™ IV (Invitrogen) and random hexamer priming (Promega Corporation, C1181). PCR amplification of IGH rearrangements from the cDNA template for HTS on the Illumina MiSeq instrument was carried out according to published protocols[32]. Each template was amplified using multiplexed IGHV primers based on the BIOMED-2 primer set in the framework, one region and one isotype-specific primer located in the first exon of the constant region[32]. These first round PCR primers also encoded half of the Illumina adaptor sequences. The first round PCR used AmpliTaq Gold (Applied 4 Biosystems) enzyme, with final primer concentrations of 3.3 μM, and the following programme: 94 °C for 7 min, 35 cycles of (94 °C for 30 s, 58 °C for 45 s, 72 °C for 120 s), and

a final extension at 72 °C for 10 min. Illumina adaptors were completed by a second PCR carried out with the Qiagen Multiplex PCR kit (Qiagen), using 0.4 µl of the first PCR product as the template in a 30 µl reaction with the following programme: 94 °C for 15 min, 12 cycles of (94 °C for 30 s, 60 °C for 45 s, 72 °C for 90 s), and a final extension at 72 °C for 10 min. Each isotype was amplified separately to decrease chimeric product generation. PCR reactions for all samples were pooled and purified by agarose gel electrophoresis and gel was extracted using the QIAquick kit (Qiagen). Libraries were sequenced as a PE300 run using a 600-cycle v3 kit.

## BCR repertoire analysis

The BCR sequence data were processed using the Immcantation toolbox (v4.0.0). Using pRESTO, raw reads were filtered according to base quality (median Phred score of ≥30). The unique molecular identifiers were annotated and a consensus sequence constructed. Forward and reverse reads were merged. Primers were trimmed and the constant region annotated[33]. Immunoglobulin gene use and sequence annotation were performed in IMGT V-QUEST[34] (for read counts see Supplementary Data. 5). The IMGT/HighV-QUEST (version 1.9.2) unique numbering provides a standardised delimitation of the framework regions (FR1-IMGT: positions 1 to 26, FR2-IMGT: 39 to 55, FR3-IMGT: 66 to 104 and FR4-IMGT: 118 to 128) and of the complementarity-determining regions: CDR1-IMGT: 27 to 38, CDR2-IMGT: 56 to 65 and CDR3-IMGT: 105 to 117.

BCR clones were assigned using the Change-O package using the single-nucleotide Hamming distance model[35]. The Alakazam package was used to analyse the BCR sequencing data for diversity estimation of CDR3 sequences; the diversity estimates were adjusted for sequencing depth by subsampling with multiple iterations[35]. Somatic hypermutation levels (including silent and non-silent mutations) per unique IGHV-D-J region per isotype were calculated over the CDR1/2 and FWR regions for each individual sample using the observed mutation function within the SHazaM package[35].

Shared IGH clones were identified based on matching V and J gene and CDR-H3 length with a minimum 85% CDR-H3 amino acid sequence identity. CDR-H3 amino acid sequence clustering was performed using CD-HIT[36] with options -c 0.85 -l 4 -S 0 -g 1 -b 1. Enrichment of select clones was determined by calculating the proportion of clones in a sample that were of interest.

## Histology and immunohistochemistry

LN tissue samples were formalin-fixed, paraffin-embedded and subsequently sectioned. Immunohistochemistry was performed on four-micron sections using the Roche Ventana BenchMark ULTRA IHC/ISH platform. Primary antibodies used were BCL6 (Leica Novocastra NCL-BCL-6-564), CD21 (Leica Novocastra NCL-CD21-2G9 clone 2G9), CD3 (Roche Ventana cat# 05278422001, clone 2GV6) and CD20 (Roche Ventana cat# 05267099001).

## Statistical methods

All statistical analyses were conducted using custom scripts in R (R Core Team, 2015).

Statistical tests were performed using Wilcoxon tests for significance (a non-parametric test of differences between distributions) and student $T$-tests for significance (a parametric test).

Differential abundance analysis was performed using the limma package[37] comparing healthy controls with disease groups with the Benjamini–Hochberg method used to control for the false discovery rate (FDR). Dysregulated gene pathways were annotated using Enrichr[38-40]. Results were visually presented as volcano plots and/or Upset plots.

Partial least-squares discriminant analysis (PLS-DA) was conducted using the *plsda()* function from the package *mixOmics*[41], a supervised method of sample discrimination whereby sample clustering is informed by group membership (here CD and UC and HC combined). The classification performance of the PLS-DA model was determined using the *perf()* function via 10 iterations of 5-fold cross-validation, with one component deemed sufficient to minimise the balanced error rate of prediction. Variable selection on components1 was conducted using the *tune()* function, with 20 antibody types selected as those most strongly contributing to the discrimination of patient groups. An AUROC curve showing the performance of a predictive model based on these 20 antibody types was generated using the *auroc()* function.

Gene set enrichment analysis (GSEA)[42] was used to identify biological pathways enriched in IBD relative to healthy controls. Briefly, a list of ranked genes, determined by signal-to-noise ratio, was generated. An enrichment score was calculated, determined by how often genes from the geneset of interest appeared at the top or the bottom of the pre-ranked set of genes, with the enrichment score representing the maximum deviation from zero. To assess statistical significance, an empirical phenotype-based permutation test was run, where a collection of enrichment scores was generated from the random assignment of phenotype to samples and used to generate a null distribution. To account for multiple testing, an FDR rate $q < 0.20$ was deemed significant. Hallmark gene sets from the Molecular Signatures Database (https://www.broadinstitute.org/gsea/msigdb) were used in the analysis.

Unsupervised clustering of differentially expressed genes was conducted using the package *ComplexHeatmap*[43] with a Euclidean distance function applied to both rows and columns of the data matrix and K means clustering performed.

## Reporting Summary

Further information on research design is available in the Nature Portfolio Reporting Summary linked to this article.

## Data availability

All data were included in the Supplementary Information or available from the authors, as are unique reagents used in this Article. The raw numbers for charts and graphs are available in the Source Data file whenever possible. The Bulk RNAseq and BCRseq data have been deposited in EGA under the Accession Number EGAD50000001335 Source data are provided with this paper.

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

## Acknowledgements

This work was funded by the Medical Research Council (Programme Grants MR/L019027/1 and MR/W018861/1). E.F.M. was supported by a Wellcome Trust (10406/Z/14/A and Beit Foundation (10406/Z/14/Z). K.G.C.S. was supported by a Wellcome Trust Senior Investigator award (200871/Z/16/Z). J.C.L. is a Lister Prize Fellow and is supported by the Francis Crick Institute, which receives core funding from Cancer Research UK (CC2219), the UK Medical Research Council (CC2219) and the Wellcome Trust (CC2219). P.K. is the recipient of a Jacquot Research Establishment Fellowship of the Royal Australasian College of Physicians Foundation, Fulbright Fellowship and NHMRC Investigator grant (2025210). L.W.U. is supported by the Erwin Schroedinger fellowship of the Austrian Science Fund (FWF J4396). We thank the Cambridge NIHR BRC Stratified Medicine Core Laboratory NGS Hub (supported by an MRC Clinical Infrastructure Award) for BCR sequencing. J.G. was supported by a Doris Duke Physician Scientist Fellowship Award (2021091), CZ Biohub Physician Scientist Scholar Award, NIH NIDDK LRP Award (2L30 DK126220), and a Stanford Maternal & Child Health Research Institute Pediatric IBD and Celiac Disease Research Award. We also thank Susanna Marquez from Yale University for help with implementing the Immcantation portal, Fan Yang from Stanford University for help with processing the post-mortem BCR datasets and Moira Finlay from the Royal Melbourne Hospital for help with histology. Graphical illustrations created with BioRender.com.

## Author contributions

Conceptualisation: P.K., W.M.R., L.W.U., E.F.M., S.D.B., K.G.C.S. and P.A.L. Data curation: P.K., L.B., D.P., L.W.U., N.M.N., S.H.T., G.M., J.L., R.S.,

S.J.S.R., B.L., D.B.B. and R.J.M.B.-R. Data analysis: P.K. Funding acquisition: S.R., J.G., S.D.B., A.S., L.W.U., E.F.M., P.A.L. and K.G.C.S. Writing—original draft: P.K and W.M.R. Writing—review & editing: P.K., W.M.R., L.B., S.H.T., S.D.B., G.M., J.L., R.S., S.J.S.R., B.L., D.B.B., L.W.U., J.C.L., E.F.M., K.G.C.S. and P.A.L. All authors have read and approved the final version of the manuscript.

## Competing interests

The authors declare no competing interests.
