## [Transparent Peer Review file · Nature Communications]

Disease-specific B cell clones are shared between patients with Crohn's disease

Corresponding Author: Dr Prasanti Kotagiri

Version 0:

Reviewer comments:

Reviewer #1

(Remarks to the Author)

The authors evaluated the expressed IgH BCR repertoire in peripheral B cells and lymph nodes of patients with Crohn Disease (CD) and performed a comparison with BCR repertoires expressed by B cells and lymph nodes in ulcerative colitis, pulmonary disease and normal controls. They found evidence of clonal expansions of B cells bearing very similar IgH sequences that were shared across many patients with CD. The findings suggest that there are a small number of antigens that are driving humoral immune responses in CD. While the data does not provide evidence of causality versus association, the authors correctly suggest that their findings could be translated into a serologic test for CD.

Following are numbered findings in this manuscript and comments regarding concerns or questions that I would suggest should be addressed in a revised version of this manuscript.

- 1) Increases in plasmablasts, serum IgA and IgG1, decreases in memory B cells
 - a. It is unclear whether the differences in plasmablasts or other B cell subsets are in percentage or absolute numbers. I am also unclear regarding the meaning of the Y axis in Figure 1 regarding B cell numbers. There should be a label that defines the units.
 - 2) Changes in the transcription patterns between 'naïve' and memory peripheral B cells in CD vs UC vs normal controls
 - a. Are the changes in transcription patterns consistent across patients?
 - b. Are there particular patterns in transcription reflective of changes in function? As presented, the transcriptional analysis offers little other than as a catalog of data. For example, is there a known inflammation B cell transcription signature that could shed light on the observed differences in transcription patterns?
 - c. Are naïve cells truly naïve, or do they contain subsets of cells that are antigen experienced? Use of the descriptor 'naïve' may be inappropriate since the CD19+IgD+CD27- cells may contain subsets that are antigen-experienced, but have not gained the markers we use to characterize memory B cells. Or do the authors have evidence that dysregulation or disease-specific changes in transcription profiles of B cells begin prior to antigen exposure? For example, some of these CD patients may have inherited disease-associated alleles of NOD2 or other disease susceptibility gene alleles.
 - 3) BCR sequences were first cloned from LN. Sequences with matching V and J genes that used identical CDR3 lengths and 85% or more CDR-H3 sequence homology were considered to be shared. Shared (present in four or more CD LNs) and disease associated (absent in control post-mortem mesenteric LN, MSLN) were further evaluated. The top 10 most frequently identified clones were common to 20 or more patients. There was enrichment for four VH4 family genes. Shared clones were also found in the peripheral blood, but at lower frequency. Among the B cell subsets evaluated, shared clones were present more frequently in plasmablasts.
 - a. Is that LN of four or more patients with CD or in four or more LN from the same patient? Please clarify.
 - b. What were the indications for surgical removal of the LN? What diagnoses, if any, did the deceased study subjects from which LN were obtained have?
 - 4) Clones enriched for IgA and IgG2
 - a. Initially it is noted that IgA and IgG1 serum (?) titers were increased. CD shared clones were more commonly found in IgA and IgG2. Explanation or discussion regarding why G1 titers were elevated but IgG2 bearing cells were more likely to be clonal expansions?
 - b. What does the Y axis refer to in Figure 1b?
 - c. Only four individuals had serum immunoglobulin levels obtained. Were these measurements performed by the clinical lab? Don't they have normal ranges? Why such a small number?
 - 5) It would appear that only H chains were sampled

- a. Is there any light chain information?
- b. The precise definition of CDR-H3 (position x to y) should be given.
- c. What if any were the characteristics of CDR-H3 that might facilitate shared sequences?
- i. The CDR3 sequences in the plasma blasts appear to be rather short, which could contribute to sequence sharing. The sequences in the LN appeared to be longer than in the plasmablasts. Admittedly what was reported was the top 10, so it is unclear whether this was a bias by looking at just 10 sequences or whether the differences in the two repertoires are significant.
- d. I note a possible increased use of JH5 and a decreased use of JH6, but the numbers are small. Decreased use of JH6 will lower CDR-H3 length distribution and reduce the contribution of tyrosine to CDR3.
- 6) Global reduction in SHM, but increased SHM among plasmablasts in CD patients
 - a. Suggests polyclonal stimulation of B cell bearing more germline repertoires, with Ag selection occurring in the plasmablasts?
- 7) Shared clones suggest the presence of common, disease-associated antigens. This leads to a possible diagnostic antibody test for CD.
 - a. Any indications that these antibodies are playing a role in disease pathogenesis?
 - b. Were patients in remission evaluated?
- 8) Supplemental Figure 3
 - a. Labels for panels a, b and c should be given.

Reviewer #2

(Remarks to the Author)

In this study by Kotagiri and colleagues, entitled “Shared hypermutated disease specific B cell clones in Crohn’s disease”, the authors leveraged B-cell receptor sequencing (BCR-Seq) technology to interrogate BCR repertoires in blood and lymphoid tissue (lymph nodes) in patients with IBD, aiming to identify IBD-specific clones with shared antigenic targets. More specifically the authors characterized CD-associated B cell clones within LNs that were shared between patients but not present in UC patients or healthy controls. Aside from LNs, such shared CD-specific repertoires were also observed in PBMCs and circulating plasmablasts. This is a very nice and well-executed study and I fully concur with the authors that more efforts should be put into unravelling B-cell pathophysiology in IBD. Strengths of the study include BCR repertoire characterization in blood (PBMCs) and tissue (draining LNs), the inclusion of multiple independent (albeit small-sized) cohorts of patients and controls, and the level of resolution gained by BCR-Seq. Limitations of the study mainly pertain to the limited biological and clinical significance put forward by this work, e.g. the absence of any functional immunological assays, the relatively small-sized cohorts introducing phenotypic heterogeneity, the poor associations to clinical phenotypes (presumably also resulting from the fact that cohorts were small in patient numbers) and the absence of intestinal mucosa-residing BCR repertoires.

Major concerns

1. While the authors performed detailed B-cell subtyping using PBMCs combined with FACS and RNA-seq techniques in both LNs and blood, characterization of mucosa-residing B-cells is lacking. This could provide more insight into shared BCR repertoires across the intestinal-systemic interface and allow discrimination between GC-derived B-cell clonal expansion and extrafollicular B-cell responses generated in GALT or Peyer’s patches, for example. Responses generated from MSLNs reflect those from migrating B-cells but do not necessarily reflect clonal responses directly generated from the intestinal tissue. Moreover, GALT is especially rich in (the diversity of) naïve B cells and therefore may have been a great target for BCR repertoire profiling.

2. As the authors touched on themselves already, this study does not provide any experimental work detailing antigenic specificity nor the neutralizing capacity of antibodies matching to the identified shared CD-associated B-cell clones. This is an important limitation, since such information would provide more insight into the translational potential of the identified common CD-associated B-cell responses into blood-based diagnostic antibody tests. The authors refer to this perspective multiple times in their manuscript, while the analysis of the exact nature of the antigens bound and/or the functional antibody capacity would be critical in my view to reliably assess the potential of the presented data for such clinical translation. In other words, the presence of shared CD-specific BCR repertoires provides structural information while this does not need to be directly related to the function of the associated antibody clones. Performing a parallel assay for antigen-specific antibody profiling such as peptide arrays or PhIP-Seq would provide more insight into the sequence-function relationships of the identified shared BCR repertoires.

3. Throughout the Results section the authors repeatedly mention the different cohorts they used for each set of analyses while it remains unclear what specific cohort was used for what type of analysis and why. Maybe the authors could better clarify this in the text, but maybe also through starting with a schematic visualization of the methodological workflow (alike Fig 2a but then study-wide and in greater detail) that they followed in their study. Furthermore, in lines 156-157 the authors state that the ‘removal of shared treatment as a potential confounding factor in the derivation of shared clones’ was enabled by a different cohort of active untreated CD patients of which PBMC-derived BCR repertoires were analyzed. The first

cohort, including patients with severe and thiopurine/biologic-treated CD (Table 1), however, was used for identification of CD-associated shared B cell clones from lymph nodes. It therefore remains undetermined whether similar CD-associated BCR repertoires would be present in LNs from active untreated CD patients. A true validation cohort for these findings would consist of a cohort of active untreated CD patients in which characterization of LN-derived BCR repertoires was performed. Discussion: "these clones are not driven by treatment" in light of these considerations I consider this an overstatement and would instead phrase that further validation is needed to confirm that these shared repertoires are indeed independent of treatment.

4. Aside from treatment, other disease-specific phenotypes such as disease location, disease behavior (e.g. inflammatory vs. fibrostenotic CD), surgical history and analysis against the presence of known serological signatures of CD/UC (e.g. ASCA/ANCA, anti-flagellin antibodies) and inflammatory disease activity (CRP, FCal, endoscopic scores, histological scores) is all lacking from the manuscript. Although I understand that cohorts were limited in size and therefore constrained in-depth phenotypic analysis, still such efforts are really important as they could drive CD-associated clones and explain the extent to which these clones are shared among included patients.

Minor points

Title

1. The title contains the word 'hypermutated' while I am not sure whether this is fully appropriate after having read the section 'Decreased SHM in IBD' in lines 187-199. Although I understand that SHM was increased when CD-associated B-cell clones (how were these defined?) were compared to HC-shared B-cell clones, the overall picture showed the opposite (decreased SHM in active IBD compared to HC). As such I would not advise to include this term in the title.

Introduction

1. line 60 "recent findings imply that..." I think this is incorrect since it is known for decades already that antibody-mediated immunity is important in IBD disease pathogenesis (e.g. ASCA/ANCA and other IBD-associated antibody reactivities).
2. line 61 "whether it is causal" this is vague in the sense that it is unclear what the hypothesized causal mechanism would be in that case, that the authors are referring to next to a breach of the mucosal barrier likely considered a collateral phenomenon.
3. lines 74-76: I doubt whether this is really true i.e. there have been studies supporting both the presence of polyclonal activation or antigen-specific response. Please rephrase.

Results

4. lines 108-112: there is apparently a great number of over- and underrepresented genes by transcriptomic characterization of naïve B-cells and memory B-cells. I would suggest the authors to highlight a few key examples of such genes that are considered biologically relevant and/or were among the top-associated transcripts.
5. lines 117-121: although it is relevant to know what the overlap was between CD- and UC-associated transcriptional signatures of B-cells, the apparent minority of non-overlapping dysregulated genes in CD and UC is not being discussed. Are there distinct clusters of genes/transcripts among these CD- and UC-specific transcriptional signatures of B-cells? It would be good to highlight such information to increase the biological significance of the presented findings.
6. lines 139-141: in their previous study (Bashford-Rogers et al, Nature 2019) they revealed an increase in B-cell clones in CD dominated by the IgA isotype. In this study, IgG+ and IgA+ fractions seem to be fairly comparable in CD-associated clones (fig 2C). Could the authors comment on this discrepancy? What are the similarities and differences?
7. lines 161-162: I do not agree with this statement (blood-based diagnostic tool), other assays would need to be performed in order to assess the potential for development into blood-based diagnostic tools.
8. line 167: 'another independent cohort' please be more clear in what specific cohort was used (see above).

Discussion

In general I find the discussion way too short in length in my opinion given the amount of data presented and the wealth of opportunities for highlighting the immunological/biological insights obtained in this study.

9. line 218: again please nuance the potential for diagnostic antibody tests since this study does not provide direct evidence of this potential (see above).

Methods

10. I am missing a section detailing the different statistical approaches that the authors used for data analysis.

Figures

In many figure panels units are missing from y-axes, please add for clarification.

Fig 1 panels f g: I would advise to color the venn diagram fields to clarify what numbers are represented by which group of individuals. This is slightly unclear at the moment.

Fig 2 panel a: I would consider to move this panel into a figure 1 and extend it to let it cover the full methodological workflow of the study. This would greatly aid readers in understanding the storyline through the results section.

Suppl fig 1: could the authors also provide some more information on the top most frequently identified clones common to most patients, e.g. have these been found in previous studies, are these 'usual suspects' in case of CD?

Reviewer #3

(Remarks to the Author)

Shared hypermutated disease specific B cell clones in Crohn's disease

Prasanti Kotagiri et al.

The authors describe Crohn's specific B-cell clones. The manuscript is very descriptive in natura and does not provide mechanical insights nor pathophysiological consideration.

It is known for decades that CD patients are characterized by circulating IgG antibodies such as ASCAs, anti-flagellin and also for example anti-pancreatic antibodies. This has been attributed to a leaky intestinal barrier and the access of bacterial antigens to the adaptive immune system of the mucosa.

The presented manuscript does not add to the existing knowledge.

It would have been of high interest to find first degree relatives of CD patients that also have increased intestinal permeability and compare their B-cell repertoire to the one of the patients. This would indicate whether the findings are truly CD specific or more related to a leaky barrier function.

In addition it would be of central importance to know whether genetic variants associated with barrier function are relevant for the B-cell pattern.

These studies would allow some functional conclusion and go beyond descriptive data.

Version 2:

Reviewer comments:

Reviewer #2

(Remarks to the Author)

The authors have satisfactorily addressed all of my previous comments. They have expanded their study by adding novel experimental data and clarified a large number of previously insufficiently clear aspects of their study. The manuscript text, including the Discussion, has also greatly improved. I would like to congratulate the authors with their work.

Reviewer #4

(Remarks to the Author)

All comments have been satisfactorily addressed except for the potential interaction between IGHJ6 usage and increased CDR3 length for the PBMCs. An updated figure and text were included with respect to the CDR3 length differences but the potential role for IGHJ6 in driving length differences between the two populations was not explore. No statistics were provided for the JH usage for 5d?

An additional cohort has been added to the study to address the concerns of Reviewer 2. This applies bulk IGH sequencing to gut mucosa samples from CD, UC and non-IBD controls. Minor comments arising from the additional IGH repertoires:

- Methods state that the sequences were aligned with IMGT VQUEST. This only allows for 50 input sequences, was it IMGT/HighV-QUEST?
- Was the same version of the IMGT Reference Directory used for the alignment of all datasets? The gene contents and gene names change between versions. As the clonotype definition includes that V gene labels there is potential that differences in the reference used could impact on shared clones.

23rd October 2023

We are grateful for the opportunity to respond to the thoughtful comments of the reviewers.

- Comments from authors in blue.

- Additions to text in purple

Reviewer #1 (expert in Ig repertoire in autoimmune disease):

The authors evaluated the expressed IgH BCR repertoire in peripheral B cells and lymph nodes of patients with Crohn Disease (CD) and performed a comparison with BCR repertoires expressed by B cells and lymph nodes in ulcerative colitis, pulmonary disease and normal controls. They found evidence of clonal expansions of B cells bearing very similar IgH sequences that were shared across many patients with CD. The findings suggest that there are a small number of antigens that are driving humoral immune responses in CD. While the data does not provide evidence of causality versus association, the authors correctly suggest that their findings could be translated into a serologic test for CD.

Following are numbered findings in this manuscript and comments regarding concerns or questions that I would suggest should be addressed in a revised version of this manuscript.

1) Increases in plasmablasts, serum IgA and IgG1, decreases in memory B cells

a. It is unclear whether the differences in plasmablasts or other B cell subsets are in percentage or absolute numbers. I am also unclear regarding the meaning of the Y axis in Figure 1 regarding B cell numbers. There should be a label that defines the units.

We have now included the relevant details in the figure axis title. The y axis label now reads as follows:

Percentage of total B cells (%)

2) Changes in the transcription patterns between 'naïve' and memory peripheral B cells in CD vs UC vs normal controls

a. Are the changes in transcription patterns consistent across patients?

We have now clustered the statistically significant differentially expressed genes of naïve and memory B cells to assess if the transcriptional patterns are consistent across patients. These figures have now been added to S1.

The following text has been added to the results.

Page 7

“We clustered on differentially expressed genes from both naive and memory B cell populations. UC and CD clustered together. In the naive B cell transcriptomic analysis, only a small proportion of IBD samples clustered with HC whilst in the memory B cell transcriptomic analysis, a small proportion of HC samples clustered with IBD (Supplementary Fig 1b and c).”

Naive B cells

Memory B cells

b. Are there particular patterns in transcription reflective of changes in function? As presented, the transcriptional analysis offers little other than as a catalog of data. For example, is there a known inflammation B cell transcription signature that could shed light on the observed differences in transcription patterns?

We have now included a transcriptomic pathway analysis.

The following text has now been added in the manuscript in the results section.

Page 6

“Highly upregulated genes (log fold change >0.5) included GBP1, 2, 4 and 5 and ITPR1 representative of interferon stimulating pathways (MSigDB Hallmark, adjusted P value 0.028) and NOD-like receptors signalling pathways (KEGG 2021 Human, adjusted P value 0.022) whilst highly downregulated genes (log fold change < -0.5) included ribosomal proteins enriching for translation (KEGG 2021 Human, adjusted P value 5.71e-17). Type I IFNs suppress protein translation, in keeping with our findings¹². A greater number of genes were dysregulated in the memory B cell transcriptome with 1609 genes upregulated, and 1243 genes downregulated. Highly upregulated genes included TRAF1, a key intracellular signalling molecule associated with auto-antibody formation and downregulation of NR4A1 and EGR3, negative regulators of self-reactive B cells, the combination potentially contributing to autoimmunity¹³⁻¹⁵.”

“We clustered on differentially expressed genes from both naïve and memory B cell populations. UC and CD clustered together. In the naïve B cell transcriptomic analysis, only a small proportion of IBD samples clustered with HC whilst in the memory B cell transcriptomic analysis, a small proportion of HC samples clustered with IBD (Supplementary Fig 1b and c). Pathway analysis of statistically significant overlapping upregulated genes in the CD and UC naïve transcriptome when compared with health (n=336 genes) revealed upregulation of the IFN alpha signalling pathway (BioCarta, adjusted P value 0.16) whilst downregulated genes (n=193) represented ribosomal pathways (KEGG 2021 human adjusted P value 1.79e-6). Pathway analysis of statistically significant overlapping upregulated genes in the CD and UC memory transcriptome when compared with health (n= 609 genes) revealed upregulation of chromatin modifying enzymes (adjusted P value 6.0e-4) whilst downregulated genes (n=497) similar to the naïve transcriptome represented ribosomal pathways (KEGG 2021 human, adjusted P value 6.0e-79). Gene set enrichment analysis identified no clear biological pathways or processes as differentially expressed between IBD (analysed as a single group) and healthy controls (Supplementary Tables 6 and 7).”

c. Are naïve cells truly naïve, or do they contain subsets of cells that are antigen experienced? Use of the descriptor ‘naïve’ may be inappropriate since the CD19+IgD+CD27- cells may contain subsets that are antigen-experienced, but have not gained the markers we use to characterize memory B cells. Or do the authors have evidence that dysregulation or disease-specific changes in transcription profiles of B cells begin prior to antigen exposure? For example, some of these CD patients may have inherited disease-associated alleles of NOD2 or other disease susceptibility gene alleles.

We agree with the reviewer that CD19+IgD+CD27- B cells may contain subsets that are antigen-experienced and thus not be fully “naïve”. We have amended the text to make this clear.

“As CD19+IgD+CD27- B cells are predominantly comprised of naïve B cells, with a small subset that are antigen-experienced, we will refer to them as naïve B cells; similarly, as CD19+IgD-CD27+CD24+CD38+ cells are predominantly comprised of memory B cells, we will refer to them as memory B cells.”

3) BCR sequences were first cloned from LN. Sequences with matching V and J genes that used identical CDR3 lengths and 85% or more CDR-H3 sequence homology were considered to be shared. Shared (present in four or more CD LNs) and disease associated (absent in control post-mortem mesenteric LN, MSLN) were further evaluated. The top 10 most frequently identified clones were common to 20 or more patients. There was enrichment for four VH4 family genes. Shared clones were also found in the peripheral blood, but at lower frequency. Among the B cell subsets evaluated, shared clones were present more frequently in plasmablasts.

a. Is that LN of four or more patients with CD or in four or more LN from the same patient? Please clarify.

It is the LN of four or more patients. We have modified the text to clarify this point:

Page 8

“To further assess the nature of these clones, we selected those that were both shared (present in the LN of four or more patients) and disease-associated (absent in control post-mortem mesenteric LN, MSLN)”

b. What were the indications for surgical removal of the LN? What diagnoses, if any, did the deceased study subjects from which LN were obtained have?

The following has been added to the supplement.

Crohn’s LN samples: Clinical information

Cohort: BCR	Disease	ID	Sex	Age	Smoker	Active disease	Immunosuppression	Disease distribution	Endoscopy/Surgical Indication	Perianal	CRP
LN	CD	BCR_1	m	40.4	yes	Yes	anti-TNF	ileocolonic	stricturing	yes	23.8
LN	CD	BCR_10	m	27.7	yes	Yes	azathioprine	colonic	stricturing	no	41.7
LN	CD	BCR_11	f	23.5	yes	Yes	anti-TNF	ileal	stricturing	yes	4.1
LN	CD	BCR_12	f	30.2	no	Yes	anti-TNF	ileal	stricturing, penetrating	no	4.9
LN	CD	BCR_13	m	62.7	no	Yes	none	ileocolonic	stricturing, penetrating	no	5.7
LN	CD	BCR_14	f	61.9	no	Yes	none	ileal	stricturing	no	8.3
LN	CD	BCR_15	f	49.9	yes	Yes	none	ileal	stricturing, penetrating	no	68
LN	CD	BCR_16	f	42.3	no	Yes	azathioprine	ileocolonic	stricturing, penetrating	yes	24.4
LN	CD	BCR_22	m	24.3	no	Yes	none	jejunal, ileal	stricturing	no	15.7
LN	CD	BCR_23	f	36.8	no	Yes	anti-TNF	colonic	non-stricturing, non-penetrating	yes	27.7
LN	CD	BCR_24	m	39.1	no	Yes	anti-TNF	jejunal, ileal	penetrating	yes	18.1
LN	CD	BCR_26	m	46.3	no	Yes	anti-TNF	ileal	stricturing	no	9.3
LN	CD	BCR_27	f	68.5	no	Yes	none	ileal	stricturing	no	4.6
LN	CD	BCR_28	m	21.2	no	Yes	azathioprine	jejunal, ileal	stricturing	no	94.5
LN	CD	BCR_29	m	29.2	no	Yes	anti-TNF	ileocolonic	stricturing, penetrating	no	148
LN	CD	BCR_30	m	49.2	no	Yes	ustekinumab	ileocolonic	penetrating	yes	168.9
LN	CD	BCR_41	m	41.5	no	Yes	none	ileocolonic	stricturing, penetrating	yes	95.5
LN	CD	BCR_46	m	30.2	no	Yes	none	ileocolonic	stricturing, penetrating	no	4.3
LN	CD	BCR_47	m	67.0	no	Yes	none	ileocolonic	stricturing	no	34.5
LN	CD	BCR_48	m	51.1	no	Yes	azathioprine	ileocolonic	stricturing, penetrating	yes	2.1
LN	CD	BCR_5	f	59.8	yes	Yes	vedolizumab	ileal	stricturing	no	5.7
LN	CD	BCR_6	f	31.7	no	Yes	none	ileocolonic	stricturing	no	7.2
LN	CD	BCR_8	f	21.7	no	Yes	none	ileocolonic	penetrating	no	30.2
LN	CD	BCR_9	m	54.3	no	Yes	none	ileocolonic	stricturing	no	3.4

Post-mortem samples: Clinical information

Donor ID	Sex	Age	Cause of death
Donor1	M	45	Trauma
Donor2	M	25	Anoxia
Donor3	M	31	Trauma
Donor4	M	46	Trauma
Donor5	M	32	Anoxia, H1N1 2009 infection
Donor6	F	32	Trauma
Donor7	F	62	Cerebrovascular accident/ Stroke
Donor8	F	43	Trauma

4) Clones enriched for IgA and IgG2

a. Initially it is noted that IgA and IgG1 serum (?) titers were increased. CD shared clones were more commonly found in IgA and IgG2. Explanation or discussion regarding why G1 titers were elevated but IgG2 bearing cells were more likely to be clonal expansions?

That is correct, IgA and IgG1 **serum** antibody levels were elevated in CD compared with healthy controls. CD shared clones were more commonly found in IgA1, IgA2 and IgG2 (Supp Fig3). To address this discrepancy, the following has been added to the discussion.

Page 11

“This study of the BCR repertoire in intestinal LNs and blood demonstrates the presence of CD-associated clones that are shared between multiple patients and are present in both LNs, PBMCs and circulating plasmablasts during active disease. These clones are not enriched in either healthy controls or during active UC to the same extent. These shared clones are more commonly found in the IgA1, IgA2 and IgG2 CD BCR repertoire despite IgA and IgG1 serum antibody levels being elevated (Supplementary Figure 4). The lack of correlation between the BCR repertoire and serum IgG1 titres may be due to differences in both cellular and immunoglobulin kinetics. “Steady state” serum IgG immunoglobulin is made predominantly by long-lived plasma cells in the bone marrow and thus may not correlate with the BCR repertoire of short-lived plasmablasts. A further consideration is differences in isotype BCR sequencing depth impacting the identification of shared clones.”

b. What does the Y axis refer to in Figure 1b?

We have changed the units to indicate this represents serum concentration in pg/ml.

We have also added the following to the methods.

Page 15

“IgA and IgG1 levels in patient serum were measured using a ProcartaPlex immunoassay kit (ThermoFisher) using 25ul of serum from each individual and run on a Luminex xMAP analyser. Raw data (MFI) were normalised to a concurrently measured 7 point standard curve according to the manufacturer’s instructions to return an absolute quantification (pg/ml). All measured values were encompassed by the standard distribution.”

c. Only four individuals had serum immunoglobulin levels obtained. Were these measurements performed by the clinical lab? Don’t they have normal ranges? Why such a small number?

Please see above. We had limited serum samples available for the analysis from cohort 1 (we have named the cohorts to make it clearer as per the response to reviewer 2). We have however since performed a measurement of total IgG in a second cohort of patients. This data has been added to the supplementary with the following text in the results.

Page 5

“In keeping with increased plasmablasts, increases in total serum IgA, IgG and IgG1 immunoglobulin titres were observed relative to health (Figure. 1b and Supplementary Figure 1a and Table 1- cohorts 1 and 2).”

5) It would appear that only H chains were sampled
a. Is there any light chain information?

Unfortunately, we lack information on the light chain, as we only conducted bulk sequencing of the heavy chain in our BCR repertoire analysis. This choice was made to maximize sequencing depth, increasing our capacity to identify public clones. Moreover, the heavy chain plays a pivotal role in binding. Typically, the light chain is crucial for the folding and positioning of the CDR H3 loop of the heavy chain, while the CDR H3 loop predominantly influences antigen binding.

b. The precise definition of CDR-H3 (position x to y) should be given.

IMGT was used to define CDR-H3. The text below has been added to the methods section.

Page 18

“The IMGT unique numbering provides a standardized delimitation of the framework regions (FR1-IMGT: positions 1 to 26, FR2-IMGT: 39 to 55, FR3-IMGT: 66 to 104 and FR4-IMGT: 118 to 128) and of the complementarity determining regions: CDR1-IMGT: 27 to 38, CDR2-IMGT: 56 to 65 and CDR3-IMGT: 105 to 117.”

c. What if any were the characteristics of CDR-H3 that might facilitate shared sequences?

i. The CDR3 sequences in the plasmablasts appear to be rather short, which could contribute to sequence sharing. The sequences in the LN appeared to be longer than in the plasmablasts. Admittedly what was reported was the top 10, so it is unclear whether this was a bias by looking at just 10 sequences or whether the differences in the two repertoires are significant.

We apologise for the confusion, the shared clones depicted in supp 2 were derived from PBMCs and not plasmablasts.

Limited clone overlap was observed among patients with CD detectable in PBMCs, as depicted in Fig 2b. We concur with the reviewer's observation that shorter sequences are more likely to be

shared. To assess this, we conducted a comparison of sequence lengths, revealing that shared clones originating from LN (comprising 2918 unique clones) tended to be slightly longer than those from PBMCs (comprising 765 unique clones), albeit not significantly. The mean length of shared LN-derived clones was 16.3 amino acids, while PBMC-derived clones averaged 15.5 amino acids. It's worth noting that shorter sequences from PBMCs were more commonly detected in CD LN, as highlighted in Supplementary Figure 2a and b, as noted by the reviewer.

This has been clarified in the text.

Page 9

“Similarly, shared clones derived from the CD PBMC repertoire (present in 2 or more individuals with CD and absent from health, n=765 clones), had greater enrichment in CD LN repertoire compared with MSLN (Supplementary Figure. 3a and b).”

d. I note a possible increased use of JH5 and a decreased use of JH6, but the numbers are small. Decreased use of JH6 will lower CDR-H3 length distribution and reduce the contribution of tyrosine to CDR3.

Analysis of J gene usage of shared clones in LN (2918 unique clones) and in PBMCs (765 unique clones) showed greater usage of J3 and J5 in shared clones derived from LN, and J4 and J6 in shared clones derived from PBMCs when compared to one another.

- 6) Global reduction in SHM, but increased SHM among plasmablasts in CD patients
a. Suggests polyclonal stimulation of B cell bearing more germline repertoires, with Ag selection occurring in the plasmablasts?

We agree with the reviewer's comment and have added this into our Discussion.

Page 13

"The global reduction in SHM is suggestive of polyclonal stimulation of B cells bearing more germline repertoires whilst the increased SHM of Crohn's associated clones is likely representative of antigen selection occurring."

- 7) Shared clones suggest the presence of common, disease-associated antigens. This leads to a possible diagnostic antibody test for CD.
a. Any indications that these antibodies are playing a role in disease pathogenesis?

This is an interesting question and we have added this into our Discussion.

Page 11

"Antibodies in patients with IBD have been previously shown to target both self-antigens and microbial components. UC commonly exhibits a high expression of self-reactive antibody pANCA²², which targets nuclear histone 1 of polymorphonuclear leukocytes. In contrast, CD is characterized by elevated levels of anti-microbial antibodies, including ASCA (anti-Saccharomyces cerevisiae antibody)²³ targeting mannan in the cell wall of baker's yeast, anti-OmpC targeting the outer-membrane protein OmpC of Escherichia coli²², anti-flagellin targeting bacterial flagellins^{24,25}, and anti-I2 targeting a component of Pseudomonas fluorescens^{18,26}.

However, it remains unclear whether all these detected antibodies play a direct role in the pathogenesis of IBD or if they merely serve as markers of an aberrant immune response. The increased frequency of antibodies against pathobionts is suggestive of a pathogenic role."

- b. Were patients in remission evaluated?

Unfortunately, we were unable to obtain MSLN samples from patients with CD in remission as surgery at this point in time is not common practice. This would have been very interesting. We instead leveraged public data available from post-mortem organ donors to act as disease controls.

- 8) Supplemental Figure 3
a. Labels for panels a, b and c should be given.

We have now included the relevant details in the figure axis.

Supp Figure 3a: Convergence of CD-associated clones in plasmablasts according to unique read count.

Supp Figure 3b: Convergence of CD-associated clones in plasmablasts according to unique clone and sub-isotype count.

Supp Figure 3c: Convergence of CD-associated clones in plasmablasts according to unique read and sub-isotype count.

Reviewer #2 (expert in IBD, IBD biomarkers, gut microbiome):

In this study by Kotagiri and colleagues, entitled “Shared hypermutated disease specific B cell clones in Crohn’s disease”, the authors leveraged B-cell receptor sequencing (BCR-Seq) technology to interrogate BCR repertoires in blood and lymphoid tissue (lymph nodes) in patients with IBD, aiming to identify IBD-specific clones with shared antigenic targets. More specifically the authors characterized CD-associated B cell clones within LNs that were shared between patients but not present in UC patients or healthy controls. Aside from LNs, such shared CD-specific repertoires were also observed in PBMCs and circulating plasmablasts. This is a very nice and well-executed study and I fully concur with the authors that more efforts should be put into unravelling B-cell pathophysiology in IBD. Strengths of the study include BCR repertoire characterization in blood (PBMCs) and tissue (draining LNs), the inclusion of multiple independent (albeit small-sized) cohorts of patients and controls, and the level of resolution gained by BCR-Seq. Limitations of the study mainly pertain to the limited biological and clinical significance put forward by this work, e.g. the absence of any functional immunological assays, the relatively small-sized cohorts introducing phenotypic heterogeneity, the poor associations to clinical phenotypes (presumably also resulting from the fact that cohorts were small in patient numbers) and the absence of intestinal mucosa-residing BCR repertoires.

Major concerns

1. While the authors performed detailed B-cell subtyping using PBMCs combined with FACS and RNA-seq techniques in both LNs and blood, characterization of mucosa-residing B-cells is lacking. This could provide more insight into shared BCR repertoires across the intestinal-systemic interface and allow discrimination between GC-derived B-cell clonal expansion and extrafollicular B-cell responses generated in GALT or Peyer’s patches, for example. Responses generated from MSLNs reflect those from migrating B-cells but do not necessarily reflect clonal responses directly generated from the intestinal tissue. Moreover, GALT is especially rich in (the diversity of) naïve B cells and therefore may have been a great target for BCR repertoire profiling.

We agree with the reviewer that this would be a very informative analysis, but it is however beyond the scope of the current study.

2. As the authors touched on themselves already, this study does not provide any experimental work detailing antigenic specificity nor the neutralizing capacity of antibodies matching to the identified shared CD-associated B-cell clones. This is an important limitation, since such information would provide more insight into the translational potential of the identified common CD-associated B-cell responses into blood-based diagnostic antibody tests. The authors refer to this perspective multiple times in their manuscript, while the analysis of the exact nature of the antigens bound and/or the functional antibody capacity would be critical in my view to reliably assess the potential of the presented data for such clinical translation. In other words, the presence of shared CD-specific BCR repertoires provides structural information while this does not need to be directly related to the function of the associated antibody clones. Performing a parallel assay for antigen-specific antibody profiling such as peptide arrays or PhIP-Seq would provide more insight into the sequence-function relationships of the identified shared BCR repertoires.

To address this, we have now assessed the serum antibody response of our patients against a range of gut commensals/pathogens to try and elucidate if there is a CD-specific antibody profile.

The following has been added to the Results.

Page 10

“Antibody binding to gut pathogens and commensals

We assessed the differential abundance of gut associated pathogen-binding antibodies in patients with CD compared with HC and those with UC combined, to obtain a disease specific response (cohort 1 and 2). Upregulation of serum antibodies targeting Klebsiella species was a dominant feature of patients with CD (Figure. 3a and Supplementary Table 9).

We used a sparse partial least-squares discriminant analysis (sPLS-DA) to determine which pathogen-associated antibodies were most informative for disease prediction. CD could be discriminated from UC and HC combined based on 20 key antibody populations selected by the model (Figure. 3b). The area under the receiver operator characteristic (AUROC) curve for patient cluster classification based on these 20 antibodies was 0.97 (97% chance of accurate cluster prediction). This is supportive but not conclusive that antibodies play a role in disease (Figure. 3c).”

The following has been added to the Discussion.

Page 12

“The serum antibody profiles of UC and CD although varied, are not sufficiently discriminative to be used solely in diagnosis¹⁸. To address this, we examined the serum antibody response of our patients against a variety of gut commensals and pathogens to uncover a potential CD-specific antibody profile. We identified a robust antibody response against *Klebsiella Pneumonia*, *Bacillus circulans*, *Lactobacillus* species, and *Enterococcus faecalis*. *Bacillus circulans* and *Lactobacillus* species are likely targeted due to their flagellar components. In a recent study using high-throughput phage-display immunoprecipitation sequencing technology to screen serum samples from patients with IBD against a panel of over 300,000 antigens, anti-flagellins antibodies were associated with ileal and fibrostenotic CD¹⁹. In addition, gut-derived *Enterococcus faecium* from IBD patients has been shown to induce colitis in genetically susceptible mice²⁰, whilst a multidrug-resistant strain of *Klebsiella* was associated with exacerbation and severity of IBD, in addition to causing intestinal inflammation in colitis-prone mice²¹.

Antibodies in patients with IBD have been previously shown to target both self-antigens and microbial components. UC commonly exhibits a high expression of self-reactive antibody pANCA²², which targets nuclear histone 1 of polymorphonuclear leukocytes. In contrast, CD is characterized by elevated levels of anti-microbial antibodies, including ASCA (anti-*Saccharomyces cerevisiae* antibody)²³ targeting mannan in the cell wall of baker's yeast, anti-OmpC targeting the outer-membrane protein OmpC of *Escherichia coli*²², anti-flagellin targeting bacterial flagellins^{24,25}, and anti-I2 targeting a component of *Pseudomonas fluorescens*^{18,26}. Our work further expands the microbial targets of CD. However, it remains unclear whether all these detected antibodies play a direct role in the pathogenesis of IBD or if they merely serve as markers of an aberrant immune response. The increased frequency of antibodies against pathobionts is suggestive of a pathogenic role.”

3. Throughout the Results section the authors repeatedly mention the different cohorts they used for each set of analyses while it remains unclear what specific cohort was used for what type of analysis and why. Maybe the authors could better clarify this in the text, but maybe also through starting with a schematic visualization of the methodological workflow (alike Fig 2a but then study-wide and in greater detail) that they followed in their study.

The following table has been added to the supplement.

	BCR repertoire: MSLN	BCR repertoire: PBMC	BCR repertoire: Plasmablasts	B cell Flow	B cell RNAseq	Serum Ig	Serum gut pathogen-specific ab
Cohort 1							
Cohort 2							
Cohort 3							

Furthermore, in lines 156-157 the authors state that the ‘removal of shared treatment as a potential confounding factor in the derivation of shared clones’ was enabled by a different cohort of active untreated CD patients of which PBMC-derived BCR repertoires were analyzed. The first cohort, including patients with severe and thiopurine/biologic-treated CD (Table 1), however, was used for identification of CD-associated shared B cell clones from lymph nodes. It therefore remains undetermined whether similar CD-associated BCR repertoires would be present in LNs from active untreated CD patients. A true validation cohort for these findings would consist of a cohort of active untreated CD patients in which characterization of LN-derived BCR repertoires was performed. Discussion: “these clones are not driven by treatment” in light of these considerations I consider this an overstatement and would instead phrase that further validation is needed to confirm that these shared repertoires are indeed independent of treatment.

We have re-phrased this statement to the following:

Page 13.

“These disease-associated clones are unlikely to be driven by treatment as we can validate their presence in patients with active CD not on immunosuppression. However, further validation is needed in a cohort of active untreated CD patients in which characterization of LN-derived BCR repertoires is performed.”

4. Aside from treatment, other disease-specific phenotypes such as disease location, disease behavior (e.g. inflammatory vs. fibrostenotic CD), surgical history and analysis against the presence of known serological signatures of CD/UC (e.g. ASCA/ANCA, anti-flagellin antibodies) and inflammatory disease activity (CRP, FCal, endoscopic scores, histological scores) is all lacking from the manuscript. Although I understand that cohorts were limited in size and therefore constrained in-depth phenotypic analysis, still such efforts are really important as they could drive CD-associated clones and explain the extent to which these clones are shared among included patients.

We agree with the reviewer of the importance of accompanying clinical data. Please see response to Reviewer 1, Q3b

Minor points

Title

1. The title contains the word ‘hypermutated’ while I am not sure whether this is fully appropriate after having read the section ‘Decreased SHM in IBD’ in lines 187-199. Although I understand that SHM was increased when CD-associated B-cell clones (how were these defined?) were compared to HC-shared B-cell clones, the overall picture showed the opposite (decreased SHM in active IBD compared to HC). As such I would not advise to include this term in the title.

We agree with the reviewers and have changed our title to the following:

“Shared disease specific B cell clones in Crohn’s disease”

Introduction

1. line 60 “recent findings imply that...” I think this is incorrect since it is known for decades already that antibody-mediated immunity is important in IBD disease pathogenesis (e.g. ASCA/ANCA and other IBD-associated antibody reactivities).

This has been rephrased.

Page 4

“IBD is classically thought to be due to aberrant T cell and innate immune responses, but mounting evidence implicates antibody-mediated immunity in disease pathogenesis.

2. line 61 “whether it is causal” this is vague in the sense that it is unclear what the hypothesized causal mechanism would be in that case, that the authors are referring to next to a breach of the mucosal barrier likely considered a collateral phenomenon.

This has been rephrased.

Page 4

“However, the nature of this involvement is unclear, i.e., whether it is causal, or the result of a breach of the mucosal barrier¹⁻⁴.”

3. lines 74-76: I doubt whether this is really true i.e. there have been studies supporting both the presence of polyclonal activation or antigen-specific response. Please rephrase.

This has been rephrased.

Page 4

“However, further work is needed to better understand the nature of the clonal B cell antigen-specific responses observed in CD.

Results

4. lines 108-112: there is apparently a great number of over- and underrepresented genes by transcriptomic characterization of naïve B-cells and memory B-cells. I would suggest the authors to highlight a few key examples of such genes that are considered biologically relevant and/or were among the top-associated transcripts.

We agree with the reviewer. Please see response to Reviewer 1, Q2

5. lines 117-121: although it is relevant to know what the overlap was between CD- and UC-associated transcriptional signatures of B-cells, the apparent minority of non-overlapping dysregulated genes in CD and UC is not being discussed. Are there distinct clusters of genes/transcripts among these CD- and UC-specific transcriptional signatures of B-cells? It would be good to highlight such information to increase the biological significance of the presented

findings.

We agree with the reviewer. Please see response to Reviewer 1, Q2

6. lines 139-141: in their previous study (Bashford-Rogers et al, Nature 2019) they revealed an increase in B-cell clones in CD dominated by the IgA isotype. In this study, IgG+ and IgA+ fractions seem to be fairly comparable in CD-associated clones (fig 2C). Could the authors comment on this discrepancy? What are the similarities and differences?

The reviewer is correct that in Bashford-Rogers et al, Nature 2019, isotype use analysis (see figure below) did not illustrate an overall increase in IgG usage in CD. In this analysis, the 'normalized' isotype use was assessed. Thus, the percentage of unique VDJ sequences per isotype were compared between disease groups. This would remove the effect of B cell clonal proliferation. However, upon assessment of clonal expansion and diversification IgG1/2 and IgA1/2 both were expanded (see figure below). This is in keeping with our findings.

7. lines 161-162: I do not agree with this statement (blood-based diagnostic tool), other assays would need to be performed in order to assess the potential for development into blood-based diagnostic tools.

We agree with the reviewer. We have changed the sentence to the following.

Page 9

“Thus, this confirms that shared antigen/s appear to drive clonal B cell responses in CD, providing an opportunity to better understand disease mechanisms.

8. line 167: ‘another independent cohort’ please be more clear in what specific cohort was used (see above).

The cohorts have now been labelled as cohort1, 2 and 3 for clarity.

Discussion

In general I find the discussion way too short in length in my opinion given the amount of data presented and the wealth of opportunities for highlighting the immunological/biological insights obtained in this study.

9. line 218: again please nuance the potential for diagnostic antibody tests since this study does not provide direct evidence of this potential (see above).

The conclusion has been moderated to the following:

“Taken together, our data not only further supports the role of pathogenic B cells in CD but also offers an opportunity, provided epitope-specific antibodies can be fully elucidated, to identify additional antigenic targets. This identification may prove valuable in aiding diagnosis, discriminating between disease endotypes and as novel treatment targets.

Methods

10. I am missing a section detailing the different statistical approaches that the authors used for data analysis.

The following has been added to the Methods section.

Page 20

Statistical methods

“All statistical analyses were conducted using custom scripts in R (R Core Team, 2015).

Statistical tests were performed using Wilcoxon tests for significance (non-parametric test of differences between distributions) and student T tests for significance (parametric test).

Differential abundance analysis was performed using the limma package³⁶ comparing healthy controls with disease groups with the Benjamini–Hochberg method used to control for the False Discovery Rate (FDR). Dysregulated gene pathways were annotated using Enrichr^{37–39}. Results were visually presented as volcano plots and/or Upset plots.

Partial least-squares discriminant analysis (PLS-DA) was conducted using the *plsda()* function from the package *mixOmics*⁴⁰ a supervised method of sample discrimination whereby sample clustering is informed by group membership (here CD and UC and HC combined). The classification performance of the PLS-DA model was determined using the *perf()* function via 10 iterations of 5-fold cross-validation, with one component deemed sufficient to minimize the balanced error rate of prediction. Variable selection on components1 was conducted using the *tune()* function, with 20 antibody types selected as those most strongly contributing to discrimination of patient groups. An AUROC curve showing the performance of a predictive model based on these 20 antibody types was generated using the *auroc()* function.

Gene set enrichment analysis (GSEA)⁴¹ was used to identify biological pathways enriched in IBD relative to healthy controls. Briefly, a list of ranked genes, determined by Signal-To-Noise ratio, was generated. An enrichment score was calculated, determined by how often genes from the geneset of interest appeared at the top or the bottom of the pre-ranked set of genes with the enrichment score representing the maximum deviation from zero. To assess statistical significance, an empirical phenotype- based permutation test was run, where a collection of

enrichment scores was generated from the random assignment of phenotype to samples and used to generate a null distribution. To account for multiple testing, an FDR rate $q < 0.20$ was deemed significant. Hallmark gene sets from the Molecular Signatures Database (<https://www.broadinstitute.org/gsea/msigdb>) were used in analysis.

Unsupervised clustering of differentially expressed genes was conducted using the package *ComplexHeatmap*⁴² with a Euclidean distance function applied to both rows and columns of the data matrix and K means clustering performed.”

Figures

In many figure panels units are missing from y-axes, please add for clarification.

This has been amended. Please see response to Reviewer 1, 1a, 4b and 8a.

Fig 1 panels f g: I would advise to color the venn diagram fields to clarify what numbers are represented by which group of individuals. This is slightly unclear at the moment.

We have replaced the venn diagrams with Upset plots to increase clarity.

Fig 2 panel a: I would consider to move this panel into a figure 1 and extend it to let it cover the full methodological workflow of the study. This would greatly aid readers in understanding the storyline through the results section.

We agree with the reviewer that clarification of the cohorts is required. Please see response to Q3.

Suppl fig 1: could the authors also provide some more information on the top most frequently identified clones common to most patients, e.g. have these been found in previous studies, are these ‘usual suspects’ in case of CD?

We agree with the reviewer that this would be very informative. Unfortunately, such information is not available. There are very few published studies studying the BCR repertoire in CD and our

work is the first to look at public clones. There currently is no publicly available BCR repertoire sequences of ASCA- specific or Flagella-specific B cells to interrogate.

Reviewer #3 (expert in IBD, gastrointestinal disorders):

Shared hypermutated disease specific B cell clones in Crohn's disease

Prasanti Kotagiri et al.

The authors describe Crohn's specific B-cell clones. The manuscript is very descriptive in natura and does not provide mechanical insights nor pathophysiological consideration.

It is known for decades that CD patients are characterized by circulating IgG antibodies such as ASCAs, anti-flagellin and also for example anti-pancreatic antibodies. This has been attributed to a leaky intestinal barrier and the access of bacterial antigens to the adaptive immune system of the mucosa.

The presented manuscript does not add to the existing knowledge.

It would have been of high interest to find first degree relatives of CD patients that also have increased intestinal permeability and compare their B-cell repertoire to the one of the patients. This would indicate whether the findings are truly CD specific or more related to a leaky barrier function.

We agree with the reviewer that assessing first degree relatives would have been of great interest. However, we were unable to obtain LN samples from this group. To address the impact of leaky barrier on the BCR repertoire we included UC in our comparisons.

In addition it would be of central importance to know whether genetic variants associated with barrier function are relevant for the B-cell pattern.

We agree with the reviewer that understanding genetics variants and how this this relates to barrier function would be relevant. However, such a study would need to be order of magnitudes larger for it to yield meaningful results. A BCR repertoire study of MSLN in >1000 patients was beyond the scope of this project.

These studies would allow some functional conclusion and go beyond descriptive data.

To address this, we have now assessed the serum antibody response of our patients against a range of gut commensals/pathogens to try and elucidate if there is a CD-specific antibody profile. Please see Reviewer 2, Q2.

15th June 2024

We are grateful for the opportunity to respond to the thoughtful comments of the reviewers.

- Comments from authors in blue.

- Additions to text in purple

Reviewer #1 (expert in Ig repertoire in autoimmune disease):

The authors evaluated the expressed IgH BCR repertoire in peripheral B cells and lymph nodes of patients with Crohn Disease (CD) and performed a comparison with BCR repertoires expressed by B cells and lymph nodes in ulcerative colitis, pulmonary disease and normal controls. They found evidence of clonal expansions of B cells bearing very similar IgH sequences that were shared across many patients with CD. The findings suggest that there are a small number of antigens that are driving humoral immune responses in CD. While the data does not provide evidence of causality versus association, the authors correctly suggest that their findings could be translated into a serologic test for CD.

Following are numbered findings in this manuscript and comments regarding concerns or questions that I would suggest should be addressed in a revised version of this manuscript.

1) Increases in plasmablasts, serum IgA and IgG1, decreases in memory B cells

a. It is unclear whether the differences in plasmablasts or other B cell subsets are in percentage or absolute numbers. I am also unclear regarding the meaning of the Y axis in Figure 1 regarding B cell numbers. There should be a label that defines the units.

We have now included the relevant details in the figure axis title. The y axis label now reads as follows:

Percentage of total B cells (%)

2) Changes in the transcription patterns between 'naïve' and memory peripheral B cells in CD vs UC vs normal controls

a. Are the changes in transcription patterns consistent across patients?

We have now clustered the statistically significant differentially expressed genes of naïve and memory B cells to assess if the transcriptional patterns are consistent across patients. These figures have now been added to S1.

The following text has been added to the results.

Page 7

“To further investigate this, we clustered IBD patients and HCs using differentially expressed genes from both the naive and memory B cell populations. Broadly, UC and CD patients clustered together. Naïve B cell transcriptomes of only a few IBD samples clustered with HC, and memory B cell transcriptomes were also distinct, with only a few HC samples clustered with IBD (Supplementary Fig 1c and d).

Naive B cells

Memory B cells

b. Are there particular patterns in transcription reflective of changes in function? As presented, the transcriptional analysis offers little other than as a catalog of data. For example, is there a known inflammation B cell transcription signature that could shed light on the observed differences in transcription patterns?

We have now included a transcriptomic pathway analysis.

The following text has now been added in the manuscript in the results section.

Page 6

“Highly upregulated genes (log fold change >0.5) included GBP1, 2, 4 and 5 and ITPR1 representative of interferon stimulating pathways (MSigDB Hallmark, adjusted P value 0.028) and NOD-like receptors signalling pathways (KEGG 2021 Human, adjusted P value 0.022) whilst highly downregulated genes (log fold change < -0.5) included ribosomal proteins enriching for translation (KEGG 2021 Human, adjusted P value 5.71e-17). Type I IFNs suppress protein translation, in keeping with our findings¹². A greater number of genes were dysregulated in the memory B cell transcriptome with 1,609 genes upregulated and 1,243 genes downregulated (Supplementary Table 4). Highly upregulated genes included TRAF1, a key intracellular signalling molecule associated with auto-antibody formation, and downregulated genes included NR4A1 and EGR3, both negative regulators of self-reactive B cells, the combination potentially contributing to autoimmunity”

“To further investigate this, we clustered IBD patients and HCs using differentially expressed genes from both the naïve and memory B cell populations. Broadly, UC and CD patients clustered together. Naïve B cell transcriptomes of only a few IBD samples clustered with HC, and memory B cell transcriptomes were also distinct, with only a few HC samples clustered with IBD (Supplementary Fig 1c and d). Pathway analysis of statistically significant overlapping upregulated genes in the CD and UC naïve transcriptome when compared with health (n=336 genes) revealed upregulation of the IFN alpha signalling pathway (BioCarta, adjusted P value 0.16), whilst downregulated genes (n=193) represented ribosomal pathways (KEGG 2021 human adjusted P value 1.79e-6). Pathway analysis of statistically significant overlapping upregulated genes in the CD and UC memory transcriptome when compared with health (n= 609 genes) revealed upregulation of chromatin modifying enzymes (adjusted P value 6.0e-4), whilst downregulated genes (n=497) were enriched for ribosomal pathways (KEGG 2021 human, adjusted P value 6.0e-79), similar to the findings in naïve B cells.”

c. Are naïve cells truly naïve, or do they contain subsets of cells that are antigen experienced? Use of the descriptor ‘naïve’ may be inappropriate since the CD19+IgD+CD27- cells may contain subsets that are antigen-experienced, but have not gained the markers we use to characterize memory B cells. Or do the authors have evidence that dysregulation or disease-specific changes in transcription profiles of B cells begin prior to antigen exposure? For example, some of these CD patients may have inherited disease-associated alleles of NOD2 or other disease susceptibility gene alleles.

We agree with the reviewer that CD19+IgD+CD27- B cells may contain subsets that are antigen-experienced and thus not be fully “naïve”. We have amended the text to make this clear.

“Since CD19+IgD+CD27- B cells are mostly naïve B cells with few antigen-experienced ones, we refer to them as naïve B cells. Similarly, since CD19+IgD-CD27+CD24+CD38+ cells are mainly memory B cells, we refer to them as memory B cells.

3) BCR sequences were first cloned from LN. Sequences with matching V and J genes that used identical CDR3 lengths and 85% or more CDR-H3 sequence homology were considered to be shared. Shared (present in four or more CD LNs) and disease associated (absent in control post-mortem mesenteric LN, MSLN) were further evaluated. The top 10 most frequently identified clones were common to 20 or more patients. There was enrichment for four VH4 family genes. Shared clones were also found in the peripheral blood, but at lower frequency. Among the B cell subsets evaluated, shared clones were present more frequently in plasmablasts.

a. Is that LN of two or more patients with CD or in two or more LN from the same patient? Please clarify.

It is the LN of two or more patients. We have modified the text to clarify this point:

“To further assess the nature of these clones, we selected those that were both shared and disease-associated (absent in control post-mortem mesenteric LN, MSLN)¹⁶ (Figure. 2a). Using this method, we identified 12,108 unique clones that were present in at least two individuals with CD and that were absent in the post-mortem MSLN samples.

We added an additional analysis where we refined the number of Crohn’s specific clones as detailed below:

Page 10

“We further refined the number of CD-associated clones derived from LNs by focusing only on the top 3,000 expanded clones per person, hypothesizing that expanded clones in the setting of active disease are more likely to be relevant to pathogenesis. Selecting those that were present in 2 or more individuals and absent in the post-mortem samples reduced the number of CD-associated LN clones to 4,698 from 12,108 clones.”

b. What were the indications for surgical removal of the LN? What diagnoses, if any, did the deceased study subjects from which LN were obtained have?

The following has been added to the supplement.

Crohn’s LN samples: Clinical information

Cohort: BCR	Disease	ID	Sex	Age	Smoker	Active disease	Immunosuppression	Disease distribution	Endoscopy/Surgical Indication	Perianal	CRP
LN	CD	BCR_1	m	40.4	yes	Yes	anti-TNF	ileocolonic	stricturing	yes	23.8
LN	CD	BCR_10	m	27.7	yes	Yes	azathioprine	colonic	stricturing	no	41.7
LN	CD	BCR_11	f	23.5	yes	Yes	anti-TNF	ileal	stricturing	yes	4.1
LN	CD	BCR_12	f	30.2	no	Yes	anti-TNF	ileal	stricturing, penetrating	no	4.9
LN	CD	BCR_13	m	62.7	no	Yes	none	ileocolonic	stricturing, penetrating	no	5.7
LN	CD	BCR_14	f	61.9	no	Yes	none	ileal	stricturing	no	8.3
LN	CD	BCR_15	f	49.9	yes	Yes	none	ileal	stricturing, penetrating	no	68
LN	CD	BCR_16	f	42.3	no	Yes	azathioprine	ileocolonic	stricturing, penetrating	yes	24.4
LN	CD	BCR_22	m	24.3	no	Yes	none	jejunal, ileal	stricturing	no	15.7
LN	CD	BCR_23	f	36.8	no	Yes	anti-TNF	colonic	non-stricturing, non-penetrating	yes	27.7
LN	CD	BCR_24	m	39.1	no	Yes	anti-TNF	jejunal, ileal	penetrating	yes	18.1
LN	CD	BCR_26	m	46.3	no	Yes	anti-TNF	ileal	stricturing	no	9.3
LN	CD	BCR_27	f	68.5	no	Yes	none	ileal	stricturing	no	4.6
LN	CD	BCR_28	m	21.2	no	Yes	azathioprine	jejunal, ileal	stricturing	no	94.5
LN	CD	BCR_29	m	29.2	no	Yes	anti-TNF	ileocolonic	stricturing, penetrating	no	148
LN	CD	BCR_30	m	49.2	no	Yes	ustekinumab	ileocolonic	penetrating	yes	168.9
LN	CD	BCR_41	m	41.5	no	Yes	none	ileocolonic	stricturing, penetrating	yes	95.5
LN	CD	BCR_46	m	30.2	no	Yes	none	ileocolonic	stricturing, penetrating	no	4.3
LN	CD	BCR_47	m	67.0	no	Yes	none	ileocolonic	stricturing	no	34.5
LN	CD	BCR_48	m	51.1	no	Yes	azathioprine	ileocolonic	stricturing, penetrating	yes	2.1
LN	CD	BCR_5	f	59.8	yes	Yes	vedolizumab	ileal	stricturing	no	5.7
LN	CD	BCR_6	f	31.7	no	Yes	none	ileocolonic	stricturing	no	7.2
LN	CD	BCR_8	f	21.7	no	Yes	none	ileocolonic	penetrating	no	30.2
LN	CD	BCR_9	m	54.3	no	Yes	none	ileocolonic	stricturing	no	3.4

Post-mortem samples: Clinical information

Donor ID	Sex	Age	Cause of death
Donor1	M	45	Trauma
Donor2	M	25	Anoxia
Donor3	M	31	Trauma
Donor4	M	46	Trauma
Donor5	M	32	Anoxia, H1N1 2009 infection
Donor6	F	32	Trauma
Donor7	F	62	Cerebrovascular accident/ Stroke
Donor8	F	43	Trauma

4) Clones enriched for IgA and IgG2

a. Initially it is noted that IgA and IgG1 serum (?) titers were increased. CD shared clones were more commonly found in IgA and IgG2. Explanation or discussion regarding why G1 titers were elevated but IgG2 bearing cells were more likely to be clonal expansions?

That is correct, IgA and IgG1 **serum** antibody levels were elevated in CD compared with healthy controls. CD shared clones were more commonly found in IgM, IgA1 and IgG2 (Supp Fig3F).

To investigate this discrepancy, we assessed enrichment of CD-associated clones within PBMCs according to **isotype** and in a new cohort of individuals where samples were obtained from the gut mucosa. We found that in the PBMC cohort, CD shared clones were more commonly found in IGHM, IGHA1/A2, and IGHG1/2 isotypes. We were not able to further separate sub-isotypes due to the primers used in this cohort. When investigating enrichment of CD-associated clones within gut mucosa. There was significant enrichment in IGHA2 with a trend towards significance in IGHA1, IGHG1, 2, and 3.

The following has been added to the results:

Page 9:

PBMCs:

“We also observed significantly more convergence of CD-associated clones in CD when analyzed by isotype, particularly with higher frequencies in the IGHM, IGHA1/A2, and IGHG1/2 isotypes (**Supplementary Figure. 3a**).”

Page 13-14:

Gut mucosa:

“Assessing for CD-associated clones in these samples, revealed greater enrichment of CD-associated clones in the repertoire of patients with CD compared to HC or patients with UC using both the expansive and refined list of clones refined list of clones with significant enrichment in the IGHM and IGHA2 and a trend towards enrichment in IGHA1 and the IGHG classes”

The following has been added to the discussion.

Page 15

“CD-associated clones were more commonly found in the IgA1, IgA2 and IgG2 CD BCR repertoire despite IgA and IgG1 serum antibody levels being elevated. The lack of correlation between the BCR repertoire and serum IgG1 titres may be due to differences in both cellular and

immunoglobulin kinetics. “Steady state” serum IgG immunoglobulin is made predominantly by long-lived plasma cells in the bone marrow and thus may not correlate with the BCR repertoire of short-lived plasmablasts. A further consideration is differences in isotype BCR sequencing depth impacting the identification of shared clones.”

b. What does the Y axis refer to in Figure 1b?

We have changed the units to indicate this represents serum concentration in pg/ml.

We have also added the following to the methods.

Page 18

“IgA and IgG1 levels in patient serum were measured using a ProcartaPlex immunoassay kit (ThermoFisher) using 25ul of serum from each individual and run on a Luminex xMAP analyser. Raw data (MFI) were normalised to a concurrently measured 7 point standard curve according to the manufacturer’s instructions to return an absolute quantification (pg/ml). All measured values were encompassed by the standard distribution.”

c. Only four individuals had serum immunoglobulin levels obtained. Were these measurements performed by the clinical lab? Don’t they have normal ranges? Why such a small number?

Please see above. We had limited serum samples available for the analysis from cohort 1 (we have named the cohorts to make it clearer as per the response to reviewer 2). We have however since performed a measurement of total IgG in a second cohort of patients. This data has been added to the supplementary with the following text in the results.

Page 5

“In keeping with increased plasmablast numbers, increases in total serum IgA, IgG and IgG1 titres were observed relative to HCs (Figure. 1b, Supplementary Figure 1b and Table 1- cohorts 1 and 2).”

5) It would appear that only H chains were sampled
a. Is there any light chain information?

Unfortunately, we lack information on the light chain, as we only conducted bulk sequencing of the heavy chain in our BCR repertoire analysis. This choice was made to maximize sequencing depth, increasing our capacity to identify public clones. Moreover, the heavy chain plays a pivotal role in binding. Typically, the light chain is crucial for the folding and positioning of the CDR H3 loop of the heavy chain, while the CDR H3 loop predominantly influences antigen binding.

b. The precise definition of CDR-H3 (position x to y) should be given.

IMGT was used to define CDR-H3. The text below has been added to the methods section.

Page 22

“The IMGT unique numbering provides a standardized delimitation of the framework regions (FR1-IMGT: positions 1 to 26, FR2-IMGT: 39 to 55, FR3-IMGT: 66 to 104 and FR4-IMGT: 118 to 128) and of the complementarity determining regions: CDR1-IMGT: 27 to 38, CDR2-IMGT: 56 to 65 and CDR3-IMGT: 105 to 117.”

c. What if any were the characteristics of CDR-H3 that might facilitate shared sequences?

i. The CDR3 sequences in the plasmablasts appear to be rather short, which could contribute to sequence sharing. The sequences in the LN appeared to be longer than in the plasmablasts. Admittedly what was reported was the top 10, so it is unclear whether this was a bias by looking at just 10 sequences or whether the differences in the two repertoires are significant.

We apologise for the confusion, the shared clones depicted in supp 2 were derived from PBMCs and not plasmablasts. We concur with the reviewer's observation that shorter sequences are more likely to be shared. In addition, shorter sequences derived from PBMCs more commonly detected in CD LN, as highlighted in Supplementary Figure 3c.

To assess length differences amongst shared clones in LN and PBMCs, we conducted a comparison of sequence lengths, revealing that shared clones originating from LN (comprising 12,108 unique clones) were significantly longer than those from PBMCs (comprising 765 unique clones). The mean length of shared LN-derived clones was 16.5 amino acids, while PBMC-derived clones averaged 15.5 amino acids.

This text has been modified as follows.

Page 9

“Similarly, shared clones derived from the CD PBMC repertoire (present in 2 or more individuals with CD and absent from health, n=765 clones), had greater enrichment in CD LN repertoire compared with MSLN (**Supplementary Figure. 3b and c**). A comparison of sequence lengths revealed that shared clones originating from LN were significantly longer with a mean length of 16.5 amino acids compared with those derived from PBMCs with a mean length of 15.5 amino acids (**Supplementary Figure. 3d**).”

d. I note a possible increased use of JH5 and a decreased use of JH6, but the numbers are small. Decreased use of JH6 will lower CDR-H3 length distribution and reduce the contribution of tyrosine to CDR3.

Analysis of J gene usage of shared clones in LN (12,108 unique clones) and in PBMCs (765 unique clones) showed greater usage of J3 and J5 in shared clones derived from LN, and J4 and J6 in shared clones derived from PBMCs when compared to one another.

6) Global reduction in SHM, but increased SHM among plasmablasts in CD patients
 a. Suggests polyclonal stimulation of B cell bearing more germline repertoires, with Ag selection occurring in the plasmablasts?

We agree with the reviewer’s comment and have added this into our Discussion.

Page 15

“The global reduction in SHM is suggestive of polyclonal stimulation of B cells bearing more germline repertoires whilst the increased SHM of Crohn’s associated clones is likely representative of antigen selection occurring.”

7) Shared clones suggest the presence of common, disease-associated antigens. This leads to a possible diagnostic antibody test for CD.

a. Any indications that these antibodies are playing a role in disease pathogenesis?

This is an interesting question and we have added this into our Discussion.

Page 16

“Antibodies in patients with IBD have been previously shown to target both self-antigens and microbial components. UC commonly exhibits a high expression of self-reactive antibody pANCA²², which targets nuclear histone 1 of polymorphonuclear leukocytes. In contrast, CD is characterized by elevated levels of anti-microbial antibodies, including ASCA (anti-Saccharomyces cerevisiae antibody)²³ targeting mannan in the cell wall of baker's yeast, anti-OmpC targeting the outer-membrane protein OmpC of Escherichia coli²², anti-flagellin targeting bacterial flagellins^{24,25}, and anti-I2 targeting a component of Pseudomonas fluorescens^{18,26}.

However, it remains unclear whether all these detected antibodies play a direct role in the pathogenesis of IBD or if they merely serve as markers of an aberrant immune response. The increased frequency of antibodies against pathobionts is suggestive of a pathogenic role.”

b. Were patients in remission evaluated?

Unfortunately, we were unable to obtain MSLN samples from patients with CD in remission as surgery at this point in time is not common practice. This would have been very interesting. We instead leveraged public data available from post-mortem organ donors to act as disease controls.

8) Supplemental Figure 3

a. Labels for panels a, b and c should be given.

We have now included the relevant details in the figure axis.

Reviewer #2 (expert in IBD, IBD biomarkers, gut microbiome):

In this study by Kotagiri and colleagues, entitled “Shared hypermutated disease specific B cell clones in Crohn’s disease”, the authors leveraged B-cell receptor sequencing (BCR-Seq) technology to interrogate BCR repertoires in blood and lymphoid tissue (lymph nodes) in patients with IBD, aiming to identify IBD-specific clones with shared antigenic targets. More specifically the authors characterized CD-associated B cell clones within LNs that were shared between patients but not present in UC patients or healthy controls. Aside from LNs, such shared CD-specific repertoires were also observed in PBMCs and circulating plasmablasts. This is a very nice and well-executed study and I fully concur with the authors that more efforts should be put into unravelling B-cell pathophysiology in IBD. Strengths of the study include BCR repertoire characterization in blood (PBMCs) and tissue (draining LNs), the inclusion of multiple independent (albeit small-sized) cohorts of patients and controls, and the level of resolution gained by BCR-Seq. Limitations of the study mainly pertain to the limited biological and clinical significance put forward by this work, e.g. the absence of any functional immunological assays, the relatively small-sized cohorts introducing phenotypic heterogeneity, the poor associations to clinical phenotypes (presumably also resulting from the fact that cohorts were small in patient numbers) and the absence of intestinal mucosa-residing BCR repertoires.

Major concerns

1. While the authors performed detailed B-cell subtyping using PBMCs combined with FACS and RNA-seq techniques in both LNs and blood, characterization of mucosa-residing B-cells is lacking. This could provide more insight into shared BCR repertoires across the intestinal-systemic interface and allow discrimination between GC-derived B-cell clonal expansion and extrafollicular B-cell responses generated in GALT or Peyer’s patches, for example. Responses generated from MSLNs reflect those from migrating B-cells but do not necessarily reflect clonal responses directly generated from the intestinal tissue. Moreover, GALT is especially rich in (the diversity of) naïve B cells and therefore may have been a great target for BCR repertoire profiling.

To address this, we have now assessed the BCR repertoire in the gut mucosa of patients with CD, UC and non-IBD controls. Samples in IBD were not only obtained from regions of active inflammation but also from areas that appeared healthy endoscopically.

The following has been added to the Results.

Page 12:

“Intestinal BCR Repertoire Analysis Reveals Enrichment of Crohn's Disease-Associated Clones

To gain a deeper understanding of the intestinal-systemic interface, we investigated the bulk BCR repertoire in intestinal mucosa during active disease (cohort 4). We compared the repertoires obtained from areas of active inflammation (CD n=27, UC n=19) and matched less inflamed samples from the same subjects (CD n=28, UC n=19) with non-IBD uninflamed controls (n=12) (Figure. 5a and Supplementary Table. 6 and 8).

In light of elevated serum IgA and G observed in IBD, we wanted to assess if there were differences in sub-isotype expression. We assessed the proportion of unique B cell clones within IgA and G subclasses, counting each unique VDJ region only once to ensure results were not

skewed by the differential mRNA content of B cell subsets. This revealed no significant difference in sub-isotype use between disease groups (**Supplementary Figure. 4a and b**).

SHM changes were consistent with findings in circulating plasmablasts with a global reduction in the class-switched isotypes including IGHA1 and 2, and IGHG1, 2 and 3 (**Figure. 5b**). This finding was most pronounced in UC as observed in the circulating plasmablasts (**Figure 3a**). There was no significant difference when comparing paired inflamed and uninfamed regions, indicating a global aberrancy in B cells in the mucosa of patients with IBD independent of inflammatory state (**Figure. 5b**). We next examined the contribution of various VH genes to the repertoire. Compared with non-IBD controls, both UC and CD showed decreased IGHV4-61 use and increased IGHV1-8 use across non-switched and class-switched isotypes. Inflamed and uninfamed regions showed similar V gene usage profiles (**Figure. 5c and Supplementary Figure. 4c**).

Assessing for CD-associated clones in these samples revealed greater enrichment of CD-associated clones in the repertoire of patients with CD compared to HC or patients with UC, using both the expansive and refined list of clones with significant enrichment in the IGHM and IGHA2, as well as a trend towards enrichment in IGHA1 and the IGHG classes (**Figure 5d and Supplementary Figure. 4d and e**). There was no greater increase in clonal convergence when comparing the inflamed and uninfamed regions in patients with CD or when comparing different regions of inflamed bowel (**Supplementary Figure. 4f-g**). In keeping with the findings in circulating plasmablasts, CD-associated clones detected in CD gut mucosa had increased rates of SHM compared to clones that were deemed non-CD associated, which suggests affinity maturation (**Figure 5e**).”

2. As the authors touched on themselves already, this study does not provide any experimental work detailing antigenic specificity nor the neutralizing capacity of antibodies matching to the identified shared CD-associated B-cell clones. This is an important limitation, since such information would provide more insight into the translational potential of the identified common CD-associated B-cell responses into blood-based diagnostic antibody tests. The authors refer to this perspective multiple times in their manuscript, while the analysis of the exact nature of the antigens bound and/or the functional antibody capacity would be critical in my view to reliably assess the potential of the presented data for such clinical translation. In other words, the presence of shared CD-specific BCR repertoires provides structural information while this does not need to be directly related to the function of the associated antibody clones. Performing a parallel assay for antigen-specific antibody profiling such as peptide arrays or PhIP-Seq would provide more insight into the sequence-function relationships of the identified shared BCR repertoires.

To address this, we have now assessed the serum antibody response of our patients against a range of gut commensals/pathogens to try and elucidate if there is a CD-specific antibody profile.

The following has been added to the Results.

Page 10

“Antibody binding to gut pathogens and commensals

We assessed the differential abundance of gut associated pathogen-binding antibodies in patients with CD compared with HC and those with UC combined, to obtain a disease specific response

(cohort 1 and 2). Upregulation of serum antibodies targeting *Klebsiella* species was a dominant feature of patients with CD (Figure. 3a and Supplementary Table 9).

We used a sparse partial least-squares discriminant analysis (sPLS-DA) to determine which pathogen-associated antibodies were most informative for disease prediction. CD could be discriminated from UC and HC combined based on 20 key antibody populations selected by the model (Figure. 3b). The area under the receiver operator characteristic (AUROC) curve for patient cluster classification based on these 20 antibodies was 0.97 (97% chance of accurate cluster prediction). This is supportive but not conclusive that antibodies play a role in disease (Figure. 3c).”

The following has been added to the Discussion.

Page 12

“In keeping with the presence of CD-associated clones, we uncovered a CD-specific antibody profile. We identified a robust antibody response against *Klebsiella* Pneumonia, *Bacillus circulans*, *Lactobacillus* species, and *Enterococcus faecalis* in patients with CD. *Bacillus circulans* and *Lactobacillus* species are likely targeted due to their flagellar components. Anti-flagellins antibodies have previously been associated with ileal and fibrostenotic CD¹⁸. In addition, gut-derived *Enterococcus faecium* from IBD patients has been shown to induce colitis in genetically susceptible mice¹⁹, whilst *Klebsiella* has been associated with exacerbations and increased severity

of IBD, in addition to causing intestinal inflammation in colitis-prone mice²⁰. Antibodies in patients with IBD have been previously shown to target both self-antigens and microbial components. UC commonly exhibits a high expression of self-reactive antibody pANCA²¹, which targets nuclear histone 1 of polymorphonuclear leukocytes. In contrast, CD is characterized by elevated levels of anti-microbial antibodies, including ASCA (anti-Saccharomyces cerevisiae antibody)²² targeting mannan in the cell wall of baker's yeast, anti-OmpC targeting the outer-membrane protein OmpC of Escherichia coli²¹, anti-flagellin targeting bacterial flagellins^{23,24}, and anti-I2 targeting a component of Pseudomonas fluorescens^{25,26}. Our work further expands the microbial targets of CD. However, it remains unclear whether all these detected antibodies play a direct role in the pathogenesis of IBD or if they merely serve as markers of an aberrant immune response. The increased frequency of antibodies against pathobionts is suggestive of a pathogenic role. Going forward, the generation of monoclonal antibodies from convergent IGH sequences would allow their further functional characterization of the identified CD-associated clones.”

3. Throughout the Results section the authors repeatedly mention the different cohorts they used for each set of analyses while it remains unclear what specific cohort was used for what type of analysis and why. Maybe the authors could better clarify this in the text, but maybe also through starting with a schematic visualization of the methodological workflow (alike Fig 2a but then study-wide and in greater detail) that they followed in their study.

The following schema has been added to the supplement Figure 1a.

The following table has been added to the supplement.

	B cell Flow	B cell RNAseq	Serum Ig	Serum gut pathogen specific antibodies	BCR repertoire: MSLN	BCR repertoire: PBMC	BCR repertoire: Plasmablasts	BCR repertoire: Gut mucosa
Cohort 1								
Cohort 2								
Cohort 3								
Cohort 4								

Furthermore, in lines 156-157 the authors state that the ‘removal of shared treatment as a potential confounding factor in the derivation of shared clones’ was enabled by a different cohort

of active untreated CD patients of which PBMC-derived BCR repertoires were analyzed. The first cohort, including patients with severe and thiopurine/biologic-treated CD (Table 1), however, was used for identification of CD-associated shared B cell clones from lymph nodes. It therefore remains undetermined whether similar CD-associated BCR repertoires would be present in LNs from active untreated CD patients. A true validation cohort for these findings would consist of a cohort of active untreated CD patients in which characterization of LN-derived BCR repertoires was performed. Discussion: “these clones are not driven by treatment” in light of these considerations I consider this an overstatement and would instead phrase that further validation is needed to confirm that these shared repertoires are indeed independent of treatment.

We have re-phrased this statement to the following:

Page 15.

“These disease-associated clones are unlikely to be driven by treatment as we can validate their presence in patients with active CD not on immunosuppression. However, further validation is needed in a cohort of active untreated CD patients in which characterization of LN-derived BCR repertoires is performed.”

4. Aside from treatment, other disease-specific phenotypes such as disease location, disease behavior (e.g. inflammatory vs. fibrostenotic CD), surgical history and analysis against the presence of known serological signatures of CD/UC (e.g. ASCA/ANCA, anti-flagellin antibodies) and inflammatory disease activity (CRP, FCal, endoscopic scores, histological scores) is all lacking from the manuscript. Although I understand that cohorts were limited in size and therefore constrained in-depth phenotypic analysis, still such efforts are really important as they could drive CD-associated clones and explain the extent to which these clones are shared among included patients.

We agree with the reviewer of the importance of accompanying clinical data. Please see response to Reviewer 1, Q3b

In addition, the following clinical data has been included for the gut samples (Supplementary Table 10).

Sample Number	IBD Diagnosis	GI Tissue	Inflammation	Age	Sex	Medication
1	UC	Colon	Non-inflamed	48	Male	AntiTNF
2	UC	Colon	Non-inflamed	37	Female	Vedolizumab
3	UC	Colon	Non-inflamed	21	Female	AntiTNF
4	UC	Colon	Inflamed	21	Female	AntiTNF
5	UC	Colon	Non-inflamed	77	Male	Ustekinumab
6	UC	Colon	Inflamed	77	Male	Ustekinumab
7	UC	Colon	Inflamed	25	Female	AntiTNF
8	UC	Colon	Non-inflamed	35	Male	AntiTNF
9	UC	Colon	Inflamed	28	Female	Vedolizumab
10	CD	Colon	Non-inflamed	32	Female	Vedolizumab
11	CD	Colon	Inflamed	32	Female	Vedolizumab

12	UC	Colon	Non-inflamed	22	Male	Mesalamine
13	UC	Colon	Inflamed	22	Male	Mesalamine
14	UC	Colon	Non-inflamed	37	Male	Mesalamine
15	UC	Colon	Inflamed	37	Male	Mesalamine
16	CD	Colon	Non-inflamed	31	Male	AntiTNF
17	CD	Colon	Inflamed	31	Male	AntiTNF
18	UC	Colon	Inflamed	50	Male	Vedolizumab
19	UC	Colon	Non-inflamed	35	Male	Mesalamine
20	UC	Colon	Inflamed	35	Male	Mesalamine
21	CD	Colon	Non-inflamed	61	Male	None
22	CD	Colon	Inflamed	61	Male	None
23	CD	Colon	Inflamed	61	Male	None
24	UC	Colon	Inflamed	29	Female	Mesalamine
25	UC	Colon	Inflamed	29	Female	Mesalamine
26	CD	Colon	Non-inflamed	28	Male	Ustekinumab
27	CD	Colon	Inflamed	28	Male	Ustekinumab
28	UC	Colon	Non-inflamed	35	Female	Vedolizumab
29	UC	Colon	Inflamed	35	Female	Vedolizumab
30	CD	Colon	Non-inflamed	35	Male	AntiTNF
31	CD	Colon	Inflamed	35	Male	AntiTNF
32	CD	Ileum	Non-inflamed	53	Male	None
33	CD	Ileum	Inflamed	53	Male	None
34	CD	Colon	Non-inflamed	53	Male	None
35	CD	Colon	Inflamed	53	Male	None
36	CD	Ileum	Non-inflamed	52	Female	None
37	UC	Colon	Non-inflamed	21	Female	Vedolizumab
38	UC	Colon	Inflamed	21	Female	Vedolizumab
39	CD	Colon	Non-inflamed	25	Female	AntiTNF
40	CD	Colon	Inflamed	25	Female	AntiTNF
41	CD	Ileum	Non-inflamed	27	Female	AntiTNF
42	CD	Ileum	Inflamed	27	Female	AntiTNF
43	CD	Colon	Non-inflamed	27	Female	AntiTNF
44	CD	Colon	Inflamed	27	Female	AntiTNF
45	UC	Colon	Non-inflamed	53	Male	Vedolizumab
46	UC	Colon	Inflamed	53	Male	Vedolizumab
47	UC	Colon	Non-inflamed	45	Female	AntiTNF
48	UC	Colon	Inflamed	45	Female	AntiTNF
49	UC	Colon	Non-inflamed	57	Male	Mesalamine
50	UC	Colon	Non-inflamed	57	Male	Mesalamine
51	CD	Colon	Non-inflamed	60	Male	AntiTNF
52	UC	Colon	Non-inflamed	35	Male	Mesalamine
53	UC	Colon	Inflamed	35	Male	Mesalamine
54	CD	Colon	Non-inflamed	32	Male	Vedolizumab

55	CD	Colon	Inflamed	32	Male	Vedolizumab
56	UC	Colon	Non-inflamed	69	Female	AntiTNF
57	UC	Colon	Inflamed	69	Female	AntiTNF
58	Non-IBD Control	Ileum	Non-inflamed	62	Male	None
59	Non-IBD Control	Ileum	Non-inflamed	62	Male	None
60	CD	Colon	Non-inflamed	72	Female	Vedolizumab
61	CD	Colon	Inflamed	72	Female	Vedolizumab
62	UC	Colon	Non-inflamed	41	Female	AntiTNF
63	UC	Colon	Inflamed	41	Female	AntiTNF
64	CD	Ileum	Non-inflamed	22	Female	AntiTNF
65	CD	Ileum	Inflamed	22	Female	AntiTNF
66	CD	Colon	Non-inflamed	22	Female	AntiTNF
67	CD	Colon	Inflamed	22	Female	AntiTNF
68	UC	Colon	Inflamed	38	Female	AntiTNF
69	CD	Ileum	Non-inflamed	36	Male	AntiTNF
70	CD	Colon	Non-inflamed	36	Male	AntiTNF
71	CD	Colon	Inflamed	36	Male	AntiTNF
72	UC	Colon	Non-inflamed	36	Male	AntiTNF
73	CD	Colon	Inflamed	24	Male	Vedolizumab
74	CD	Colon	Non-inflamed	24	Male	Vedolizumab
75	CD	Ileum	Non-inflamed	32	Male	Vedolizumab
76	CD	Ileum	Inflamed	32	Male	Vedolizumab
77	CD	Colon	Non-inflamed	65	Male	Vedolizumab
78	CD	Colon	Inflamed	65	Male	Vedolizumab
79	CD	Ileum	Non-inflamed	75	Male	None
80	CD	Ileum	Inflamed	75	Male	None
81	CD	Colon	Non-inflamed	75	Male	None
82	CD	Colon	Inflamed	75	Male	None
83	CD	Ileum	Non-inflamed	21	Male	None
84	CD	Ileum	Inflamed	21	Male	None
85	CD	Colon	Non-inflamed	21	Male	None
86	CD	Colon	Inflamed	21	Male	None
87	CD	Ileum	Non-inflamed	55	Male	None
88	CD	Ileum	Inflamed	55	Male	None
89	CD	Colon	Inflamed	55	Male	None
90	Non-IBD Control	Ileum	Non-inflamed	66	Male	None
91	Non-IBD Control	Colon	Non-inflamed	66	Male	None
92	UC	Colon	Non-inflamed	58	Female	Vedolizumab
93	UC	Colon	Inflamed	58	Female	Vedolizumab
94	Non-IBD Control	Ileum	Non-inflamed	46	Female	None
95	Non-IBD Control	Colon	Non-inflamed	46	Female	None
96	Non-IBD Control	Ileum	Non-inflamed	56	Female	None
97	Non-IBD Control	Colon	Non-inflamed	56	Female	None

98	Non-IBD Control	Ileum	Non-inflamed	57	Male	None
99	Non-IBD Control	Colon	Non-inflamed	57	Male	None
100	Non-IBD Control	Ileum	Non-inflamed	47	Female	None
101	Non-IBD Control	Colon	Non-inflamed	47	Female	None
102	CD	Colon	Non-inflamed	43	Male	Vedolizumab
103	CD	Colon	Inflamed	43	Male	Vedolizumab
104	CD	Colon	Non-inflamed	72	Female	Vedolizumab
105	CD	Colon	Inflamed	72	Female	Vedolizumab

Minor points

Title

1. The title contains the word ‘hypermutated’ while I am not sure whether this is fully appropriate after having read the section ‘Decreased SHM in IBD’ in lines 187-199. Although I understand that SHM was increased when CD-associated B-cell clones (how were these defined?) were compared to HC-shared B-cell clones, the overall picture showed the opposite (decreased SHM in active IBD compared to HC). As such I would not advise to include this term in the title.

We agree with the reviewers and have changed our title to the following:

“Shared disease specific B cell clones in Crohn’s disease”

Introduction

1. line 60 “recent findings imply that...” I think this is incorrect since it is known for decades already that antibody-mediated immunity is important in IBD disease pathogenesis (e.g. ASCA/ANCA and other IBD-associated antibody reactivities).

This has been rephrased.

Page 4

“IBD is classically thought to be due to aberrant T cell and innate immune responses, but mounting evidence implicates antibody-mediated immunity in disease pathogenesis.

2. line 61 “whether it is causal” this is vague in the sense that it is unclear what the hypothesized causal mechanism would be in that case, that the authors are referring to next to a breach of the mucosal barrier likely considered a collateral phenomenon.

This has been rephrased.

Page 4

“However, the nature of this B cell involvement is unclear, especially whether it is causal or simply the result of a breach in the mucosal barrier.”

3. lines 74-76: I doubt whether this is really true i.e. there have been studies supporting both the presence of polyclonal activation or antigen-specific response. Please rephrase.

This has been rephrased.

Page 4

“However, further work is needed to better understand the precise role in disease of the clonal B cell antigen-specific responses observed in CD”

Results

4. lines 108-112: there is apparently a great number of over- and underrepresented genes by transcriptomic characterization of naïve B-cells and memory B-cells. I would suggest the authors to highlight a few key examples of such genes that are considered biologically relevant and/or were among the top-associated transcripts.

We agree with the reviewer. Please see response to Reviewer 1, Q2

5. lines 117-121: although it is relevant to know what the overlap was between CD- and UC-associated transcriptional signatures of B-cells, the apparent minority of non-overlapping dysregulated genes in CD and UC is not being discussed. Are there distinct clusters of genes/transcripts among these CD- and UC-specific transcriptional signatures of B-cells? It would be good to highlight such information to increase the biological significance of the presented findings.

We agree with the reviewer. Please see response to Reviewer 1, Q2

6. lines 139-141: in their previous study (Bashford-Rogers et al, Nature 2019) they revealed an increase in B-cell clones in CD dominated by the IgA isotype. In this study, IgG+ and IgA+ fractions seem to be fairly comparable in CD-associated clones (fig 2C). Could the authors comment on this discrepancy? What are the similarities and differences?

The reviewer is correct that in Bashford-Rogers et al, Nature 2019, isotype use analysis (see figure below) did not illustrate an overall increase in IgG usage in CD. In this analysis, the ‘normalized’ isotype use was assessed. Thus, the percentage of unique VDJ sequences per isotype were compared between disease groups. This would remove the effect of B cell clonal proliferation. However, upon assessment of clonal expansion and diversification IgG1/2 and IgA1/2 both were expanded (see figure below). This is in keeping with our findings.

7. lines 161-162: I do not agree with this statement (blood-based diagnostic tool), other assays would need to be performed in order to assess the potential for development into blood-based diagnostic tools.

We agree with the reviewer. We have changed the sentence to the following.

Page 9

“The enrichment of clones underscored the reproducibility of these convergent B cell responses in CD disease mechanisms”

8. line 167: ‘another independent cohort’ please be more clear in what specific cohort was used (see above).

The cohorts have now been labelled as cohort1, 2, 3 and 4 for clarity.

Discussion

In general I find the discussion way too short in length in my opinion given the amount of data presented and the wealth of opportunities for highlighting the immunological/biological insights obtained in this study.

9. line 218: again please nuance the potential for diagnostic antibody tests since this study does not provide direct evidence of this potential (see above).

The conclusion has been moderated to the following:

“Taken together, our data not only further supports the role of pathogenic B cells in CD but also offers an opportunity, provided epitope-specific antibodies can be fully elucidated, to identify additional antigenic targets. This identification may prove valuable in aiding diagnosis, discriminating between disease endotypes and as novel treatment targets.”

Methods

10. I am missing a section detailing the different statistical approaches that the authors used for data analysis.

The following has been added to the Methods section.

Statistical methods

“All statistical analyses were conducted using custom scripts in R (R Core Team, 2015).

Statistical tests were performed using Wilcoxon tests for significance (non-parametric test of differences between distributions) and student T tests for significance (parametric test).

Differential abundance analysis was performed using the *limma* package³⁶ comparing healthy controls with disease groups with the Benjamini–Hochberg method used to control for the False Discovery Rate (FDR). Dysregulated gene pathways were annotated using *Enrichr*^{37–39}. Results were visually presented as volcano plots and/or Upset plots.

Partial least-squares discriminant analysis (PLS-DA) was conducted using the *plsda()* function from the package *mixOmics*⁴⁰ a supervised method of sample discrimination whereby sample clustering is informed by group membership (here CD and UC and HC combined). The classification performance of the PLS-DA model was determined using the *perf()* function via 10 iterations of 5-fold cross-validation, with one component deemed sufficient to minimize the balanced error rate of prediction. Variable selection on components1 was conducted using the *tune()* function, with 20 antibody types selected as those most strongly contributing to discrimination of patient groups. An AUROC curve showing the performance of a predictive model based on these 20 antibody types was generated using the *auroc()* function.

Gene set enrichment analysis (GSEA)⁴¹ was used to identify biological pathways enriched in IBD relative to healthy controls. Briefly, a list of ranked genes, determined by Signal-To-Noise ratio, was generated. An enrichment score was calculated, determined by how often genes from the geneset of interest appeared at the top or the bottom of the pre-ranked set of genes with the enrichment score representing the maximum deviation from zero. To assess statistical significance, an empirical phenotype- based permutation test was run, where a collection of enrichment scores was generated from the random assignment of phenotype to samples and used to generate a null distribution. To account for multiple testing, an FDR rate $q < 0.20$ was deemed significant. Hallmark gene sets from the Molecular Signatures Database (<https://www.broadinstitute.org/gsea/msigdb>) were used in analysis.

Unsupervised clustering of differentially expressed genes was conducted using the package *ComplexHeatmap*⁴² with a Euclidean distance function applied to both rows and columns of the data matrix and K means clustering performed.”

Figures

In many figure panels units are missing from y-axes, please add for clarification.

This has been amended. Please see response to Reviewer 1, 1a, 4b and 8a.

Fig 1 panels f g: I would advise to color the venn diagram fields to clarify what numbers are represented by which group of individuals. This is slightly unclear at the moment.

We have replaced the venn diagrams with Upset plots to increase clarity.

Fig 2 panel a: I would consider to move this panel into a figure 1 and extend it to let it cover the full methodological workflow of the study. This would greatly aid readers in understanding the storyline through the results section.

We agree with the reviewer that clarification of the cohorts is required. Please see response to Q3.

Suppl fig 1: could the authors also provide some more information on the top most frequently identified clones common to most patients, e.g. have these been found in previous studies, are these 'usual suspects' in case of CD?

We agree with the reviewer that this would be very informative. Unfortunately, such information is not available. There are very few published studies studying the BCR repertoire in CD and our work is the first to look at public clones. There currently is no publicly available BCR repertoire sequences of ASCA- specific or Flagella-specific B cells to interrogate.

Reviewer #3 (expert in IBD, gastrointestinal disorders):

Shared hypermutated disease specific B cell clones in Crohn's disease

Prasanti Kotagiri et al.

The authors describe Crohn's specific B-cell clones. The manuscript is very descriptive in natura and does not provide mechanical insights nor pathophysiological consideration.

It is known for decades that CD patients are characterized by circulating IgG antibodies such as ASCAs, anti-flagellin and also for example anti-pancreatic antibodies. This has been attributed to a leaky intestinal barrier and the access of bacterial antigens to the adaptive immune system of the mucosa.

The presented manuscript does not add to the existing knowledge.

It would have been of high interest to find first degree relatives of CD patients that also have increased intestinal permeability and compare their B-cell repertoire to the one of the patients. This would indicate whether the findings are truly CD specific or more related to a leaky barrier function.

We agree with the reviewer that assessing first degree relatives would have been of great interest. However, we were unable to obtain LN samples from this group. To address the impact of leaky barrier on the BCR repertoire we included UC in our comparisons.

In addition it would be of central importance to know whether genetic variants associated with barrier function are relevant for the B-cell pattern.

We agree with the reviewer that understanding genetics variants and how this this relates to barrier function would be relevant. However, such a study would need to be order of magnitudes larger for it to yield meaningful results. A BCR repertoire study of MSLN in >1000 patients was beyond the scope of this project.

These studies would allow some functional conclusion and go beyond descriptive data.

To address this, we have now assessed the serum antibody response of our patients against a range of gut commensals/pathogens to try and elucidate if there is a CD-specific antibody profile. Please see Reviewer 2, Q2.

18th September 2024

We are grateful for the opportunity to respond to the thoughtful comments of the reviewers.

- Comments from authors in blue.

- Additions to text in purple

Reviewer #2 (Remarks to the Author):

The authors have satisfactorily addressed all of my previous comments. They have expanded their study by adding novel experimental data and clarified a large number of previously insufficiently clear aspects of their study. The manuscript text, including the Discussion, has also greatly improved. I would like to congratulate the authors with their work.

We thank the reviewers for these comments.

Reviewer #4 (Remarks to the Author):

All comments have been satisfactorily addressed except for the potential interaction between IGHJ6 usage and increased CDR3 length for the PBMCs. An updated figure and text were included with respect to the CDR3 length differences but the potential role for IGHJ6 in driving length differences between the two populations was not explore. No statistics were provided for the JH usage for 5d?

We thank the reviewer for this comment and have now performed a Fisher's exact test followed by a pairwise analysis with Bonferroni correction. This analysis has been added to the supplement (Supplementary Figure. 3d)

"A comparison of sequence lengths revealed that shared clones originating from LN were significantly longer with a mean length of 16.5 amino acids compared with those derived from PBMCs with a mean length of 15.5 amino acids (**Supplementary Figure. 3c**). A comparison of J gene frequency in shared clones derived from LN versus PBMCs yielded a Pearson's Chi-squared test $P = 0.0004998$, indicative of differences in J gene usage across the two sites. A post-hoc Bonferroni-corrected pairwise analysis revealed significant proportional differences between IGHJ4 (PBMCs 84%, LN 77%) versus IGHJ5 (PBMCs 16%, LN 23%) with a corrected P value = 0.004, and IGHJ5 (PBMCs 41%, LN 54%) versus IGHJ6 (PBMCs 59%, LN 46%) with a corrected P value = 0.006 (**Supplementary Figure. 3d**)."

Supplementary Figure. 3d

An additional cohort has been added to the study to address the concerns of Reviewer 2. This applies bulk IGH sequencing to gut mucosa samples from CD, UC and non-IBD controls.

Minor comments arising from the additional IGH repertoires:

- Methods state that the sequences were aligned with IMGT VQUEST. This only allows for 50 input sequences, was it IMGT/HighV-QUEST?

We apologise for the confusion, IMGT/HighV-QUEST was used to define CDR-H3. The text in the methods section has been modified.

“The IMGT/HighV-QUEST (version 1.9.2) unique numbering provides a standardized delimitation of the framework regions (FR1-IMGT: positions 1 to 26, FR2-IMGT: 39 to 55, FR3-IMGT: 66 to 104 and FR4-IMGT: 118 to 128) and of the complementarity determining regions: CDR1-IMGT: 27 to 38, CDR2-IMGT: 56 to 65 and CDR3-IMGT: 105 to 117.”

- Was the same version of the IMGT Reference Directory used for the alignment of all datasets? The gene contents and gene names change between versions. As the clonotype definition includes that V gene labels there is potential that differences in the reference used could impact on shared clones.

We agree with the reviewer. Version 1.9.2 was used. Please see above.